# Sustainable Strategy Analysis: Platform Channel Configuration and Slotting Fee Design under Differentiated Quality Investment

Chunyu Li [1], Peng Xing [1,*] and Yanting Li [2]

[1] Business School, Faculty of Economics, Liaoning University, Shenyang 110136, China
[2] School of Business Administration, Northeastern University, Shenyang 110169, China
*   Correspondence: pxing@lnu.edu.cn

**Abstract:** As a medium for matching supply and demand, platforms are changing industrial structures and consumption patterns to achieve sustainable operations. The platform establishes a self-operated channel on the basis of the present agent channel, which generates new conflicts and pollution problems. Considering the competition and quality investment, we investigate the platform's optimal strategies, i.e., pricing, quality investment, channel format and slotting fee contract. The result shows that the platform adopting a dual channel structure contributes to sustainable operations because it can increase selling prices, sales volumes and consumers' willingness when the channel introduction cost is lower. Meanwhile, the supplier always prefers the dual-channel structure because it can increase sales volume, profits and consumer surplus. Meanwhile, contrary to the commission rate, the emergence of competition promotes quality investment and guarantees consumer satisfaction. Under the intense channel conflicts, a variable slotting fee contract (VFC) is more profitable than a unit fixed slotting fee contract (UFC) owing to alleviating the competition; otherwise, the UFC has a larger improvement effect on profits. Meanwhile, with the increase in consumer quality sensitivity, UFC gradually cannibalizes VFC and occupies the core position in the operation.

**Keywords:** sustainable strategy; self-operated channel; variable slotting fee contract; unit fixed slotting fee contract; quality investment

## 1. Introduction

In the past 20 years, various digital platforms, such as retailing platforms, APP stores and matching platforms, have sprung up and experienced explosive growth, forming a huge economic scale [1,2]. By 2019, there were 74 digital platform companies whose market capitalization was more than USD 10 billion in the global market, with a USD 8.98 trillion total value and an annual increase of 41.8% (http://www.brsn.net/xwzx/zhongwen/detail/20200605/1005000000034141591342964562638697_1.html, accessed on 19 June 2022). As a forerunner, the retailing platform plays a key role in the digital economy [1,3–6] pointed out that the emergence of various platforms promoted a central revolution in production and consumption patterns. For example, in China, the transaction volume of the online market was more than USD 1.989 trillion in 2019, and online shopping is becoming mainstream [7,8]. The new online business modes created by platforms have presented strong adaptability, which implies that online shopping is becoming a popular consumption pattern [9,10]. This has prompted more and more merchants to join online platforms to sell products directly or indirectly with different slotting fee contracts, i.e., variable, unit fixed and total fixed slotting fee contracts (i.e., VFC, UFC and TFC) [11,12]. Under the VFC, the platform takes part of the supplier's sales revenue as the platform usage. Under the UFC (TFC), the supplier pays a fixed rent to the platform per unit of product (per year). For example, Tmall.com charges 1.5–7% revenue commission; HJXMall.com charges the slotting fee by bargaining pricing, and part of the virtual product platforms

(e.g., newspaper) charges a unit fixed fee [13,14]. Guan et al. [15] show that all members care about their own profit in the supply chain; thus, an appropriate slotting fee contract can prevent suppliers from quitting the platform owing to the unfair distribution of benefits [16]. Although the platform provides a robust slotting fee contract for suppliers within the same category product, a platform contracts with thousands of suppliers, which is crucial for the platform to find a suitable slotting fee contract.

Motivated by mobile consumption, online sales are penetrating various industries and their operating modes are gradually beginning to diversify [17,18]. The platform, as the core role of online retail, attempts to establish self-operated channels in addition to providing open online direct channels for suppliers to seize the market; for instance, directly operated store (vivas) and Tmall Supermarket [19,20]. Meanwhile, some platforms, such as Pinduoduo.com, only sell products through agency channels, and they establish self-supporting channels such as Tmall.com in the future or not with the consumption upgrade. However, many platforms operate the online market by mixed channels, causing new conflicts among firms [21]. Facing channel conflicts, supply chain members try to invest in quality to maintain superiority, which enhances the market influence and weakens the negative effect of competition [22–24]. Thus, some scholars pointed out that different members all had the incentive to invest in quality to improve their bargaining power [15,24]. Quality investment has become the core way to deal with competition [24,25]. For instance, in order to effectively compete with Samsung, Apple Inc. invests to improve the quality difference with competitors and enhance its competitiveness [26]. However, the quality investment needs expensive costs. Hence, it is an important proposition for business and academics how firms effectively invest in quality to cope with market changes [27–29].

However, the existing research does not consider the scenario where platforms can encroach, nor do they involve the difference in slotting fee contracts [11,14,19,24]. In view of the research gap, we lucubrate the interaction between platforms' slotting fee contracts and new channel introductions. Specifically, under the fixed and variable entry fee contract, this paper studies the platform's channel encroachment and quality investment strategies, and explores the impact of encroachment cost and quality sensitivity on the sustainable operation of the supply chain. Based on the practice of platform economics and the aforesaid investigation, this paper specifically puts forward the following research questions: (a) What are the conditions for the platform to formulate VFC or UFC, and how does it maximize its profit? (b) Under different slotting fee contracts, how should the supplier respond to the channel introduction of the platform? (c) What is the impact of different channel structures on quality investment strategies? Which mode is more favorable? (d) How does the interaction between the platform slotting fee contracts and channel introduction affect consumer surplus?

To answer the above questions, we develop a Stackelberg game-theoretical model in a supply chain consisting of a supplier and a platform, in which the supplier directly sells products on the platform by offering a slotting fee. Based on the difference between slotting fee contracts and whether the platform should establish self-operated distribution channels, we propose four scenarios. One is the benchmark scenario of a single online agent channel composed of suppliers and platforms under VFC or UFC, as well as the scenario of a dual channel (agent and reselling channel) composed of suppliers and platforms under VFC or UFC. Meanwhile, taking channel quality investment into consideration, the optimization problems of the supply chain are constructed and solved. Comparing the equilibrium results in different cases, we analyze the influence of operational structure change of the supply chain on the equilibrium results and explore the condition of Pareto improvement. Furthermore, we obtain the optimal strategies of quality investment, channel configuration and slotting fee contracts. Finally, we discuss the optimal strategy choice of supply chain members by extending the research on endogenous slotting fee rates.

Compared to the previous literature, this paper has the following contributions in the following three aspects: (a) According to the existing literature, the interesting question arising is that a UFC and a VFC are widely applied, but most of the related research focuses

on the VFC [1,11,30,31]. Based on this research gap, we consider the main aims of sustainable cooperation between platforms and suppliers, explore the evolution of the slotting fee contract from a fixed mode to a variable mode, and obtain the internal mechanism that most platforms choose VFC and only a few choose UFC. Further, their impact on the operations of suppliers and platforms is analyzed. (b) Existing literature on supply chain encroachment mainly focuses on upstream manufacturers launching the encroachment and directly investing in channels [21,32,33], ignoring that downstream enterprises also have the motivation to initiate new channels. Motivated by the previous research [23,24], this paper captures that the platform encroaches on the supplier's online market by establishing an online self-operated channel in the context of traditional manufacturer encroachment and discusses the impact of the channel introduction cost on platform channel strategy. (c) In the field of the sustainable supply chain, most scholars pay more attention to product R&D and quality awareness of consumers [23,24], but fewer studies differentiated investment under the channel competition. In view of this, this paper explores the sustainable investment of online channels under the dynamic supply chain structure and analyzes the influence of different channel modes on investment, which fills the gap of differentiated quality investment in a competitive environment.

The remainder of this paper is as follows. In Section 2, we review the involved relevant literature. In Section 3, we describe and define the research problem. In Section 4, we construct two models based on platform channel introduction, namely, the UFC and the VFC. In Section 5, we discuss the optimal strategies of the supplier and platform. Specifically, in Section 6, we relax relevant constraints to study quality decisions and channel configuration strategies. Finally, we present the conclusions and findings in Section 7.

## 2. Literature Review

In a platform supply chain that includes one upstream supplier and one downstream platform, the platform chooses whether to introduce an own-channel and can encroach on the existing market by configuring reselling online channels under differentiated slotting fee contracts. Based on the practice of many platform operations, we investigate the encroachment and quality investment strategies. Further, from the perspective of consumer surplus, we examine the optimal online channel configuration strategy. Our work relates to the following three research aspects: platform slotting fees, channel configuration and quality investment. Next, with the proposed research goal, we review the existing literature in detail based on the three topics.

### 2.1. Platform Slotting Fee Contract

The slotting fee is a prepaid fee that the supplier must pay to the platform/retailer for its shelf. Platform slotting fee contracts are generally divided into the following three types: the TFC, UFC and VFC [12–14]. Wang et al. [13] proposed a new composite contract that combines the TFC with repurchase functions and pointed out that the composite format could provide greater benefits for stakeholders in the supply chain. Under the potential driving factors of the market size difference and product substitution, Shen [4] established a Stackelberg model to explore the effect of the TFC and VFC on channel operation. On the basis of the traditional offline agent channel, it is expanded research that retailers should provide a retail platform that directly connects merchants and consumers to grab market share and enhance brand awareness. Considering the TFC and VFC simultaneously, Shen et al. [11] found that slotting fees were not always beneficial to platforms and not always harmful to manufacturers, which depends on the substitution effect between the two retail channels. Tian et al. [34] utilized revenue-sharing contracts to make both the platform and supplier benefit, which provides testable empirical research on the relationship between different factors. Based on consumer value, Liu and Ke [35] analyzed the influence of slotting fees and pricing timing on the optimal policy of enterprises under the dual-channel format. In practice, TFC has no effect on decision-making.

The aforementioned literature mainly discusses the influence of platform slotting fees on strategic operations in different aspects. However, from the perspective of differentiated slotting fee contracts, how to make the optimal slotting fee decision is not discussed. Regarding the emerging research trend of platform retailing, the most related research to our study is Zhang et al. [14], who considered differentiated slotting fee contracts (UFC and VFC) and studied the relationship between platform contract selection and manufacturers' quality decisions. Different from Zhang et al. [14], we analyze the role of UFC and VFC from the perspective of dynamic channel structure and differentiated quality investment. Furthermore, we explore the evolution of platform slotting fee contracts with platform operation diversification and detect the basis for establishing slotting fee contracts of the platform under different channel structures.

### 2.2. Supply Chain Channel Configuration

With the rapid development of e-commerce and the improvement in consumer acceptance of online channels, suppliers (manufacturers) or retailers open up direct/indirect channels to compete with the original channel [21,32]. The literature on initial channel configuration focused on whether suppliers should add online channels and further analyzed the impact of the channel introduction on the decisions [36]. Combined with the characteristics of enterprise software, Li et al. [37] explored the optimal distribution strategies of enterprise software and proposed that channel introduction could bring the highest profit and social welfare under high unsuitable costs. Inspired by mainstream retail platforms adopting a store-in-shop strategy, Shen et al. [11] studied the interaction between retailers' sales modes and manufacturers' channel selections. The emergence of omni-channel consumers promotes retailers' multi-channel operations, resulting in channel structure diversification. Combined with the advantages (market size expansion) and disadvantages (cost, delivery time and channel competition) of online channels, Ye et al. [38] investigated whether retailers should establish online channels next to their offline physical stores. Meanwhile, based on the manufacturer and retailer configure channels, the firm's operational strategy is explored. Considering the channel introduction strategy, Nie et al. [20] investigated the impact of bilateral supply chain encroachment and determined the conditions for upstream and downstream enterprises to establish new channels in the longitudinal supply chain.

Most of the aforesaid studies focused on the change in online-offline channel structure from the supplier's perspective and investigated the impact of new channel introductions on supply chain decisions. Different from the prior channel configuration literature, this paper has the following innovations. On the one hand, from the perspective of the downstream supply chain, it explores the situation where the platform establishes an online distribution channel to encroach on the online direct selling channel, enriching the research on dual-channel operations. On the other hand, in a platform supply chain, the agency channel and the reselling channel of platforms are studied at the same time, and further, the sustainable strategies of platforms are explored, which enriches the research of the platform operation.

### 2.3. Supply Chain Quality Investment

Quality is an important measure for consumers to buy products through different online channels; thus, quality investment is the internal condition and basic power of sustainable economic development [39]. The existing studies on this topic mainly improved the investment mode. Chen and Deng [23] analyzed the impact of certification standards on a supplier's quality investment when the buyer outsourced the production process. He et al. [40] established a dynamic model to investigate the impact of reference quality and reference price on the decisions of the supply chain system. Avinadav et al. [29] constructed the framework of co-investment in quality and explored the contract design in the supply chain, resulting in the optimal contract mode of the supply chain. Chen et al. [41] adopted the following three-stage dynamic game theory to analyze the optimal mode of

quality investment: CM investment, OEM cooperation investment or OEM non-cooperation investment. Furthermore, some scholars explored the interaction between channel structure and quality investment. Chen et al. (2017) studied price and quality decisions in the supply chain, discussed the impact of channel structure change on quality investment strategy, and found that quality improvement could be achieved by introducing a new channel. Xia et al. [24] studied the service level and distribution channel decisions of two competitive supply chains and further explored the effect of service competition on the channel structure choice of the supply chain. Wang et al. [42] mainly discussed the interaction between the quality investment strategy of e-retailers and the online channel selection of suppliers and dissected the influence of marketing capabilities and investment strategies on channel selection.

The prior papers mainly focus on the production process quality, but exists a gap in the research on quality investment in the context of channel competition. Moreover, the previous papers neglected the influence of the strategic behavior of supply chain members on the quality decision. Therefore, combined with concept construction, we extend this body of literature by the game theory and explore investment strategies of suppliers and platforms under different channel structures and the core factors that influence the difference in quality investments between the supplier channel and platform channel. Furthermore, we analyze how supply chains construct operational strategies to induce suppliers and platforms to invest in quality.

A comparison of this study with the major relevant literature from the above literature review presents the limitations of the prior research and illustrates the improvement of this work on the previous research issues such as slotting fee contracts, differentiated quality investment and channel configuration. Referred the above literature review, we find that it is of great significance to study the interaction between platform channel encroachment and slotting fee design under differentiated quality investment in the context of dynamic operation. Firstly, we contribute the literature on platform slotting fee contract design through the analysis of endogenous and exogenous slotting rates, as well as obtain the design conditions of fixed and variable slotting fee contracts. Secondly, we contribute to channel configuration literature by simultaneously operating two sales models in the supply chain. Thirdly, we contribute to the quality of investment literature by considering the quality of investment of different members between two competitive channels. We believe these research results can offer valuable insights into both platforms' practice and theoretical development.

### 3. Model Setup

The platform charges the slotting fee from suppliers according to product categories and provides corresponding operational services to suppliers, such as operation planning of product sales, store design, basic market data information and a third-party payment service. The supplier directly sells its own brand product on the platform. In the process of supplier sales, the platform may open its self-operated channel to compete with suppliers under differentiated quality investment levels and selling prices. The channel formats are shown in Figure 1.

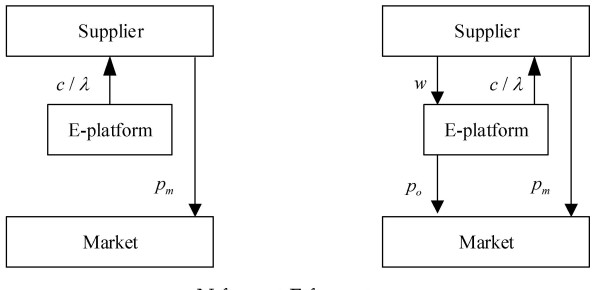

N format E format

**Figure 1.** The channel structures under different modes.

(1) Slotting fee contract

When the supplier sells products by agent channel, the platform formulates the differentiated slotting fee contract, i.e., a fixed slotting fee contract (UFC) or a variable slotting fee contract (VFC).

Under the UFC, the platform charges the unit fixed cost (CF) per unit of product as the slotting fee.

Under the VFC, the platform charges the commission rate (VR) per selling price as the slotting fee.

(2) Channel structure

The platform decides whether or not to encroach on the supplier market, and the channel structures are divided into the N format and the E format.

N format: No-encroachment, there is only an agent channel on the platform.

E format: There are two channels in the market, namely, an agent channel and a self-operated channel of the platform.

(3) Market demand

Supply chain operations are affected by channel quality investment. We assume that the quality investment level is $s_i$ and its cost is $s_i^2/2$ [33,43,44]. We take the platform's agent mode as the baseline. In the absence of encroachment, the inverse demand function is as follows: $p_i^N = a + q_i - Q_i$ (Ha et al. 2021 ($N \in \{NG, NS\}$)). In the presence of encroachment, the inverse demand function is as follows: $p_i^S = a + q_i - Q_i - \phi Q_j$ ($S \in \{SG, SS\}$) [45,46] (Table 1). The notations in this paper are listed in Table 2.

**Table 1.** A summary of main literature.

| Reference | UFC(TFC) | VFC | Encroachment Type | Quality |
|---|---|---|---|---|
| Arya et al. [32] | × | × | Manufacturer | × |
| Ha et al. [21] | × | × | Manufacturer | √ |
| Huang et al. [33] | × | × | Manufacturer | √ |
| Li et al. [22] | × | × | Retailer | √ |
| Zhang et al. [44] | × | × | Manufacturer | √ |
| Cui et al. [47] | × | × | Contract manufacturer | √ |
| Qin et al. [7] | × | Exogenous | × | √ |
| Liu et al. [1] | × | Exogenous | × | √ |
| Tian [34] | × | Exogenous | × | × |
| Shen [4] | TFC, Exogenous | Exogenous | × | × |
| Shen et al. [11] | TFC, Endogenous | Exogenous | × | × |
| Zhang et al. [14] | UFC, Endogenous | Endogenous | × | √ |
| Our work | UFC, Exogenous and Endogenous | Exogenous and Endogenous | Platform | √ |

**Table 2.** List of notations.

| Indices | Definition |
|---|---|
| $a$ | Market size. |
| $p_i, p_j$ | Selling price of the $i$ channel ($i, j = S, O; i \neq j$). |
| $Q_i, Q_j$ | Sales volume of the $i$ channel ($i, j = S, O; i \neq j$). |
| $q_i, q_j$ | Quality investment level of the $i$ channel ($i, j = S, O; i \neq j$). |
| $w$ | Wholesale price. |
| $c$ | Unit fixed cost. |
| $\phi$ | Substitution rate between agent channel and self-operated channel ($\phi \in [0, 1]$). |
| $\kappa$ | Unit commission rate (i.e., UR), and $\kappa = c/a$. |
| $\lambda$ | Variable commission rate. |
| $T$ | Platform encroachment cost. |

(4) Sequence of events

Considering the platform's encroachment decision, the sequence of events is as follows:

(a) The platform only opens up the agent channel. The supplier sets the sales volume, and then the supplier determines the quality investment level.

(b) The platform establishes the self-operated channel. The supplier first sets the wholesale price, and then the supplier and platform decide the sales volume of each channel simultaneously. Finally, the supplier and platform decide the quality investment level of each channel simultaneously.

## 4. Model Analysis

### 4.1. Model Analysis under the UFC

When the platform operates in the N format, its revenue is derived from the slotting fee submitted by the supplier. When the platform operates in the E format, it needs to order the product from the supplier, and its profit is obtained from the two channels. We consider whether the platform establishes a self-operated channel or not under the UFC and further formulate the platform agent channel model (NG mode) and platform dual-channel model (SG mode).

#### 4.1.1. NG Mode

Through paying the CF to the platform, the supplier directly sells products to consumers in the absence of encroachment. The profit functions of the supplier and platform are, respectively, as follows:

$$\max_{Q_s, q_s} \Pi_S^{NG} = (p_S - c)Q_S - C(q_S) \tag{1}$$

$$\Pi_O^{NG} = cQ_S \tag{2}$$

**Lemma 1.** *For $0 \leq \kappa \leq 1$, there is a non-negative unique equilibrium under the NG. The optimal decisions of the supplier and platform and are $Q_S^{NG} = a - c$ and $q_S^{NG} = a - c$, the optimal profits of the supplier and platform are $\Pi_S^{NG} = (a - c)^2/2$ and $\Pi_O^{NG} = c(a - c)$.*

**Corollary 1.** *The impact of the CF on decisions and profits is as follows:*
*(a) $\frac{\partial Q_S^{NG}}{\partial c} = \frac{\partial q_S^{NG}}{\partial c} < 0; \frac{\partial \Pi_S^{NG}}{\partial c} < 0$.*
*(b) If $c < a < 2c, \frac{\partial \Pi_O^{NG}}{\partial c} < 0$; if $a \geq 2c, \frac{\partial \Pi_O^{NG}}{\partial c} \geq 0$.*

**Proof.** See Appendix A. □

Corollary 1 implies that a higher CF shrinks the supplier's sales volume, leading to a lower profit, thus supplier decreases the quality investment in order to save the expense. Meanwhile, when $c < a < 2c$, the market size of the platform is small, and suppliers' revenue will rapidly decrease in the CF, which even causes the supplier to withdraw from the platform, as a result, the platform cannot obtain additional revenue. When $a > 2c$, the platform has a large traffic flow, thereby it possesses greater power in the game process, causing the profit increases in the CF.

#### 4.1.2. SG Mode

When the platform encroaches on the supplier market, the cost of the encroachment is $T$ and the supplier provides products to the platform at wholesale price $w$. The profit functions of the supplier and platform are as follows:

$$\max_{Q_S, q_S, w} \Pi_S^{SG} = (p_S - c)Q_S + wQ_O - C(q_S) \tag{3}$$

$$\max_{Q_O, q_O} \Pi_O^{SG} = (p_O - w)Q_O + cQ_S - C(q_O) - T \tag{4}$$

**Lemma 2.** *For* $0 \leq \phi^2 \leq \frac{2}{3}$, *there is a non-negative unique equilibrium under the SG. The optimal decisions the supplier and platform are* $Q_S^{SG} = \frac{a(2-\phi^2+\phi)+c(\phi^2-2)}{2-3\phi^2}$, $q_S^{SG} = \frac{a(2-\phi^2+\phi)+c(\phi^2-2)}{2-3\phi^2}$, $Q_O^{SG} = \frac{2a\phi+a-2c\phi}{2-3\phi^2}$, $q_O^{SG} = \frac{2a\phi+a-2c\phi}{2-3\phi^2}$ *and* $w^{SG} = \frac{a-a\phi^3-2a\phi^2+c\phi^3}{2-3\phi^2}$; *the profits of the supplier and platform are* $\Pi_S^{SG} = \frac{a^2((\phi^2+4\phi+3)+\kappa^2(\phi^2+2)-2\kappa(\phi^2+2\phi+2))}{4-6\phi^2}$ *and* $\Pi_O^{SG} = \frac{a^2((2\phi+1)^2-2\kappa^2(3\phi^4-10\phi^2+4)-2\kappa(3\phi^4-3\phi^3-12\phi^2+4))}{2(2-3\phi^2)^2} - T$.

**Corollary 2.** *The impact of the substitution rate and the CF on decisions and profits:*

(a) $\frac{\partial Q_S^{SG}}{\partial \phi} = \frac{\partial q_S^{SG}}{\partial \phi} > 0; \frac{\partial w^{SG}}{\partial \phi} > 0; \frac{\partial Q_O^{SG}}{\partial \phi} = \frac{\partial q_O^{SG}}{\partial \phi} > 0; \frac{\partial Q_S^{SG}}{\partial c} = \frac{\partial q_S^{SG}}{\partial c} < 0; \frac{\partial w^{SG}}{\partial c} > 0; \frac{\partial Q_O^{SG}}{\partial c} = \frac{\partial q_O^{SG}}{\partial c} < 0.$

(b) $\frac{\partial \Pi_S^{SG}}{\partial \phi} > 0; \frac{\partial \Pi_O^{SG}}{\partial \phi} < 0; \frac{\partial \Pi_S^{SG}}{\partial c} < 0.$ *If* $0 < \phi < 0.56$ *and* $\kappa > \kappa_{SG}^S, \frac{\partial \Pi_S^{SG}}{\partial c} > 0$; *if* $0 < \phi < 0.56$ *and* $0 \leq \kappa \leq \kappa_{SG}^S, \frac{\partial \Pi_S^{SG}}{\partial c} > 0$; *if* $0.56 < \phi < \phi_{SG}^C, \frac{\partial \Pi_S^{SG}}{\partial c} < 0$; *if* $\phi_{SG}^C \leq \phi \leq \phi_{SG}, \frac{\partial \Pi_S^{SG}}{\partial c} > 0.$

**Proof.** See Appendix A. □

Corollary 2 shows that the sales volumes, quality investment levels and profit of the supplier all increase with the substitution rate while a decrease in the CF. The wholesale price is positively correlated with both the substitution rate and the CF. The platform's profit is negatively correlated with the substitution rate, and its change with the CF is determined by the substitution rate and market capacity.

With fierce channel competition, the supplier would improve quality investment to seize the market and increase sales volume in order to obtain higher profits. While the supplier's profit decreases in the CF, thus the supplier will increase the wholesale price to recover profit losses and maintain its channel power. However, due to the double marginalization caused by channel competition, although the platform's quality investment increases, sales volume decrease, which lessens the platform's overall profit. When the substitution rate is weak or the market capacity is large, the supplier's wholesale price increases in the CF, but the platform revenue can be increased by obtaining additional market demand and the CF. When the substitution rate is high, the increase in the CF exacerbates channel conflicts and reduces platform revenue, which is not conducive to its operation.

*4.2. Model Analysis under the VFC*

According to the product categories of the supplier, the platform takes a portion of its selling revenue as a usage fee. We consider whether the platform establishes self-operated channels under the VFC and further construct the agency channel model (NS mode) and dual-channel model (SS mode).

4.2.1. NS Mode

When the platform does not encroach, the supplier sells products directly through the platform under the VFC. The profit functions of the supplier and platform profit are, respectively, as follows:

$$\max_{Q_S, q_S} \Pi_S^{NS} = (1-\lambda)p_S Q_S - C(q_S) \tag{5}$$

$$\Pi_O^{NS} = \lambda p_S Q_S \tag{6}$$

**Lemma 3.** *Under the NS, the sales volume and quality investment level of the supplier are* $Q_S^{NS} = \frac{a}{1+\lambda}$ *and* $q_S^{NS} = \frac{a(1-\lambda)}{1+\lambda}$; *the profits of the supplier and platform are* $\Pi_S^{NS} = \frac{a^2(1-\lambda)}{2(\lambda+1)}$ *and* $\Pi_O^{NS} = \frac{a^2\lambda}{(\lambda+1)^2}$.

**Proof.** See Appendix A. □

**Corollary 3.** *The impact of the VR on the decisions and profits is as follows:* $\frac{\partial Q_S^{NS}}{\partial \lambda} < 0$; $\frac{\partial q_S^{NS}}{\partial \lambda} < 0$; $\frac{\partial \Pi_S^{NS}}{\partial \lambda} < 0$; $\frac{\partial \Pi_O^{NS}}{\partial \lambda} > 0$.

**Proof.** See Appendix A. □

Corollary 3 proves that the sensitivity of decisions and profits to the VR is similar to that under the UFC. With the increase in the VR, the supplier decreases the quality investment level to economize on operating expenses, which results in a decrease in sales volume and the final revenue. The platform can gain additional revenue by increasing the commission rate.

4.2.2. SS Mode

In the presence of the encroachment, the platform establishes a self-operated channel and competes with the supplier. The profit functions of the supplier and platform are, respectively, as follows:

$$\max_{Q_S, q_S, w} \Pi_S^{SS} = (1 - \lambda) p_S Q_S + w Q_O - C(q_S) \tag{7}$$

$$\max_{Q_O, q_O} \Pi_O^{SS} = (p_O - w) Q_O + \lambda p_S Q_S^{SS} - C(q_O) - T \tag{8}$$

**Lemma 4.** *For* $\max\{\lambda^{SS}, 0\} \leq \lambda \leq 1$, *there exist the optimal decisions under the SS. Furthermore, the optimal decisions of the supplier and the platform are* $Q_S^{SS} = \frac{a(\phi-2)(\phi+1)}{\lambda(\phi^2-2)+3\phi^2-2}$, $Q_O^{SS} = \frac{a(\lambda+2\phi+1)}{2-\lambda(\phi^2-2)-3\phi^2}$,
$w^{SS} = \frac{a(\phi+1)\left(\lambda(\phi^2-\phi-1)+\phi^2+\phi-1\right)}{\lambda(\phi^2-2)+3\phi^2-2}$, $q_S^{SS} = \frac{a(1-\lambda)(\phi-2)(\phi+1)}{\lambda(\phi^2-2)+3\phi^2-2}$ *and* $q_O^{SS} = \frac{a(\lambda+2\phi+1)}{2-\lambda(\phi^2-2)-3\phi^2}$; *the equilibrium profits of the supplier and the platform are* $\Pi_S^{SS} = \frac{a^2(\phi+1)(\lambda(\phi-1)+\phi+3)}{2(2-\lambda(\phi^2-2)-3\phi^2)}$ *and*
$\Pi_O^{SS} = \frac{\left(a^2\left(\lambda^2(2\phi^3-2\phi^2-4\phi+1)+(2\phi+1)^2+2\lambda(3\phi^4-3\phi^3-8\phi^2+4\phi+5)\right)\right)}{2(\lambda(\phi^2-2)+3\phi^2-2)^2} - T$.

**Corollary 4.** *The effect of the substitution rate and the VR on decisions and profits is as follows:*
*(a)*① $\frac{\partial Q_S^{SS}}{\partial \phi} > 0$; $\frac{\partial q_S^{SS}}{\partial \phi} > 0$; $\frac{\partial Q_O^{SS}}{\partial \phi} > 0$; $\frac{\partial q_O^{SS}}{\partial \phi} > 0$. *If* $0 < \lambda < \lambda^0, \frac{\partial w^{SS}}{\partial \phi} < 0$; *if* $\lambda^0 \leq \lambda \leq 1, \frac{\partial w^{SS}}{\partial \phi} \geq 0$. ② $\frac{\partial Q_S^{SS}}{\partial \lambda} < 0$; $\frac{\partial q_S^{SS}}{\partial \lambda} < 0$; $\frac{\partial w^{SS}}{\partial \lambda} > 0$; $\frac{\partial Q_O^{SS}}{\partial \lambda} < 0$; $\frac{\partial q_O^{SS}}{\partial \lambda} < 0$.
*(b)* $\frac{\partial \Pi_S^{SS}}{\partial \phi} > 0$; $\frac{\partial \Pi_S^{SS}}{\partial \lambda} < 0$; $\frac{\partial \Pi_O^{SS}}{\partial \phi} > 0$. *If* $\phi \in [0, 0.467]$ *and* $0 < \lambda < \lambda^1, \frac{\partial \Pi_O^{SS}}{\partial \lambda} > 0$; *if* $\phi \in [0, 0.467]$ *and* $\lambda^1 < \lambda < 1, \frac{\partial \Pi_O^{SS}}{\partial \lambda} < 0$; *if* $\phi \in [0.467, \phi_{SG}], \frac{\partial \Pi_O^{SS}}{\partial \lambda} < 0$.

**Proof.** See Appendix A. □

Corollary 4 shows that the sales volume and quality investment of the supplier increase in the substitution rate while a decrease in the VR. If $0 < \lambda < \lambda^0$, the wholesale price decreases in the substitution rate; otherwise, it increases; meanwhile, it is negatively correlated with the VR. Therefore, the increase in the substitution rate can increase channel profits. The supplier's profit decreases in the VR. The platform's profit is affected by the substitution rate and the VR. If the two factors are both small, the platform profit increases. If the substitution rate is small and the VR is large, or if the substitution rate is large, the platform profit decreases.

As the channel competition intensifies, the supplier and platform improve their core competitiveness by improving quality investment to occupy the market and obtain additional benefits. As the VR increases, the supplier's unit revenue is too much deprived by the platform, which leads to lower quality investment and sales volume, finally reducing

the profits. However, the supplier increases the wholesale price to make up for the losses, weakening the platform's competitive advantage. Therefore, the platform has to reduce quality investment to maintain revenue, resulting in lower sales volume. When the channel competition (i.e., substitution rate) is weak and the VR is small, the supplier and the platform are in a benign relationship. As the VR increases, the platform can obtain additional benefits. Conversely, when the VR is large or the competition is fierce, there is a strong competitive relationship between the supplier and the platform, thus it leads to an increase in the wholesale price and a decrease in the platform's profit.

## 5. Comparison of Equilibrium Results

We compare combinations (NG, NS), (SG, SS), (NG, SG) and (NS, SS) in this section, obtain optimal quality investment strategy and channel structure, detect a suitable slotting fee contract, and explore the consumer surplus in different channels. We expect that the results of this study can provide guidance for enterprises. The associated proof is provided in Appendix A.

*5.1. Quality Investment*

**Proposition 1.** *Under different slotting fee contracts, comparing the supplier's quality level:*
  *(a)* ① $q_S^{NG} < q_S^{SG}$. ② $q_S^{NS} < q_S^{SS}$.
  *(b)* ① *If* $0 \leq \kappa \leq \kappa_s^0, q_S^{NS} \leq q_S^{NG}; \kappa_s^0 < \kappa \leq 1, q_S^{NS} > q_S^{NG}$. ② *For* $0 \leq \lambda \leq \lambda_s^1$, *if* $0 \leq \kappa \leq \kappa_s^1, q_S^{SG} \geq q_S^{SS}$; *if* $\kappa_s^1 < \kappa \leq 1, q_S^{SG} < q_S^{SS}$; *otherwise,* $\lambda_s^1 < \lambda \leq 1, q_S^{SG} > q_S^{SS}$.

Proposition 1 implies the impact of operational strategy changes on quality investment. The emergence of channel competition can always promote suppliers to invest in quality. Under the same channel structure, the decisions of quality investment are comprehensively affected by the substitution rate and unit commission ratio (i.e., $\kappa$ and $\lambda$). When the substitution rate is small, both channels can operate normally and the quality investment strategy is similar to that under the single channel. If the substitution rate is high, the quality investment is gradually approaching zero under the UFC. The VFC increasingly cannibalizes the UFC. When the dual channel is operated, the UFC is more conducive to maintaining the excellent quality investment with a higher UR or a smaller the VR, and vice versa.

In general, Modes NG and NS always have a disadvantage to quality investment and cannot obtain high-quality in the channel. Mode SG can always obtain a higher quality level when the fixed slotting fee is small. On the other hand, a variable slotting fee is more advantageous to stimulate quality investment. Mode SS is more favorable to quality investment when the proportion of commission is small, as shown in Figure 2.

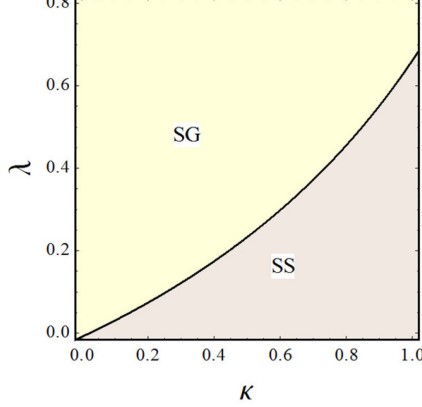

**Figure 2.** Supplier's quality investment strategy.

**Proposition 2.** *Under different slotting fee contracts, comparing the platform's quality level:* $0 < \phi < 2/3$, *if* $0 \le \kappa \le \kappa_o^1$, $q_O^{SG} \ge q_O^{SS}$; *if* $\kappa_o^1 < \kappa \le 1$, $q_O^{SG} < q_O^{SS}$. *For* $2/3 \le \phi < \sqrt{2/3}$, $q_O^{SG} < q_O^{SS}$.

Proposition 2 shows that the two channels are in a healthy state when the channel competition is small; the UFC is better for a quality investment with a lower UR; otherwise, the VFC is more beneficial to a quality investment. However, as channel competition intensifies, the small-scale platforms or the large CF cannot provide high-quality investment under the UFC, thus UFC is gradually replaced by the VFC, as shown in Figure 3.

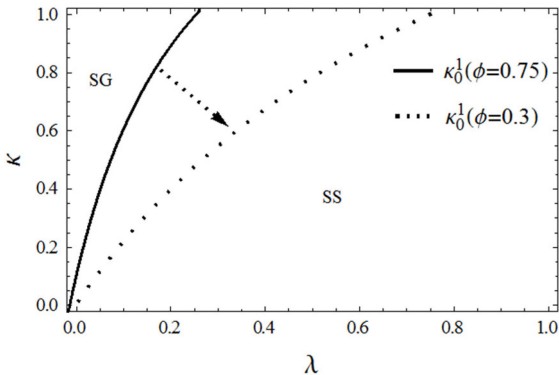

**Figure 3.** E-tailer' investment strategy.

On the whole, channel competition arouses a crisis sense in members, thus the encroachment always improves quality investment in an agent and self-operated channels, which meets the requirement of customer groups with high environmental awareness. Meanwhile, the party with a higher commission ratio (VR or UR) has a lower quality investment under each slotting fee contract. Therefore, the platform can promote high-quality operations by appropriately reducing the slotting fee rate to improve its reputation of the platform and promote consumer satisfaction.

*5.2. Channel Configuration Strategy*

**Proposition 3.** *Under the UFC, the impact of channel encroachment on the profits of supplier and platform:* (a) $\Delta\Pi_S^G > 0$. (b) $\Delta\Pi_O^G > 0$ *if* $0 \le T < T_G$; $\Delta\Pi_O^G \le 0$ *if* $T \ge T_G$.

Proposition 3 shows the channel structure preferences of the platform and the supplier. For platforms, they can gain additional benefits by opening up a self-operated channel and prefer to adopt the mixed channel model when channel introduction cost is low. When channel introduction cost is high, the self-operated channel introduction cannot bring positive profits, so the platform prefers a single-channel structure. For the supplier, the introduction of a self-operated channel can improve the sales of products, expand the market and thus increase profits, so the supplier always prefers to sell products through mixed channels.

**Proposition 4.** *Under the VFC, the impact of channel encroachment on the profits of supplier and platform:* (a) $\Delta\Pi_S^S > 0$. (b) $\Delta\Pi_O^S \ge 0$ *if* $0 \le T \le T_S$; $\Delta\Pi_S^S < 0$ *if* $T > T_S$.

Proposition 4 shows the impact of channel introduction on the supplier and platform under the VFC. However, e-retailers, its preference is affected by channel introduction cost. When the cost of channel introduction is high, the channel introduction needs to invest a lot of upfront costs, resulting in the loss of platform profit. Therefore, the platform prefers a single-channel structure. Otherwise, the introduction of a self-operated channel can bring positive income to e-retailers, and the platform prefers the mixed channel structure.

Although channel competition will lead to certain profit losses in the direct selling channel, the loss can be made up through the wholesale premium, so the supplier can always obtain extra income from expanded sales when the self-operated channel is introduced, and the supplier always prefers the mixed channel structure.

In a word, in the actual operation of platforms with large market capacities (e.g., JD.com), the cost of establishing a self-operated channel is relatively low, so the platform constructs the self-operated channel to encroach. The encroachment strategy leads to channel competition between the supplier and the platform, which can optimize quality investment and ensure the interest of consumers. Under the VFC, the profits of the platform and supplier increase, which achieves a win-win. Under the UFC, platform channel encroachment will harm the interests of the supplier. Therefore, with the development of the platform, the UFC is gradually being replaced by the VFC.

In a word, suppliers always prefer mixed-channel operation, as shown in Figure 4a. For the platform, it prefers mixed-channel operation when the cost of channel introduction is low; otherwise, a single-channel structure is better, as shown in Figure 4b. Therefore, for the platform with large market capacity (e.g., Suning.com and JD.com, accessed on 6 October 2022), most platforms take mixed channel operations. For small-sized platforms, a single-channel structure is more conducive to sustainable operation.

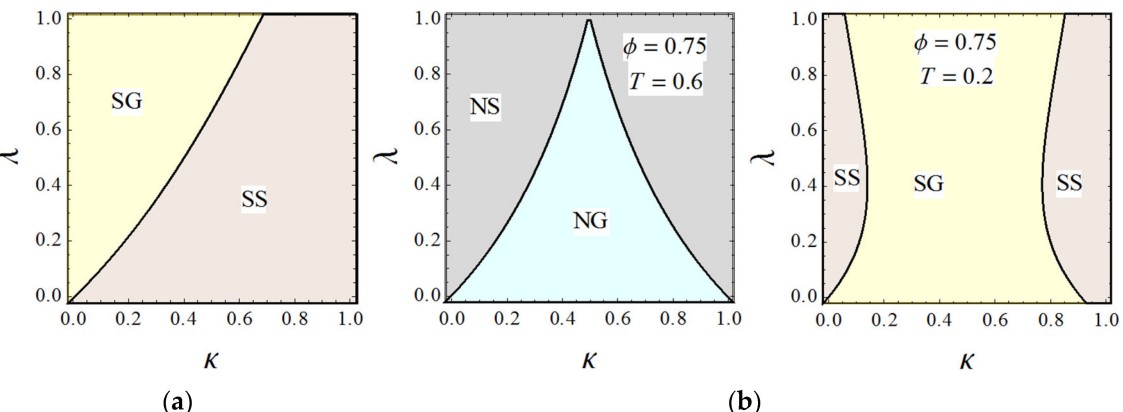

**Figure 4.** Mode preference of supplier and platform. (**a**) The supplier's Mode preference (**b**) The platform's Mode preference.

*5.3. Slotting Fee Contract*

**Proposition 5.** *In the absence of the channel introduction, the optimal slotting fee contract: (a)* $\Delta\Pi_S^N > 0$ *if* $\kappa_1 < \kappa \leq 1$; $\Delta\Pi_S^N \leq 0$ *if* $0 < \kappa \leq \kappa_1$. *(b)* $\Delta\Pi_O^N \leq 0$ *if* $\kappa'_1 \leq \kappa \leq \kappa'_2$; $\Delta\Pi_O^N > 0$ *if* $0 \leq \kappa < \kappa'_1$ *or* $\kappa'_2 < \kappa \leq 1$.

Proposition 5 implies that adopting the VFC is better for the supplier when the UR is larger than the VR; otherwise, the supplier prefers the UFC. For the platform, adopting the VFC is beneficial when $0 \leq \kappa < \kappa'_1$ or $\kappa'_2 < \kappa \leq 1$; otherwise, the UFC is superior. In brief, a small UR or a low VR can merely ensure one's benefit. Specifically, when the UR (VF) is lower, the platform and supplier have the same preferences; otherwise, the supplier's choice is contrary to the platform's choice, as shown in Figure 5.

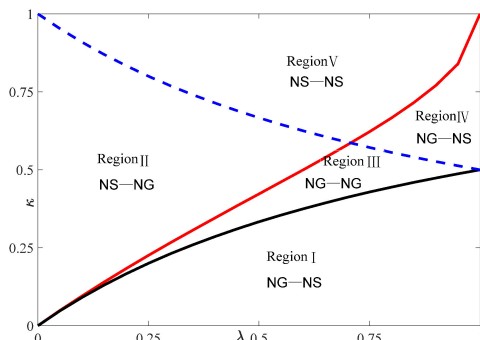

**Figure 5.** Equilibrium slotting fee contract in Mode N.

**Proposition 6.** *In the presence of the channel introduction, the optimal slotting fee contract: (a)* $\Delta\Pi_S^S \leq 0$ *if* $0 < \kappa < \kappa''$; *otherwise,* $\Delta\Pi_S^S > 0$ *if* $\kappa_1'' < \kappa < 1$ *or* $\kappa_1'' < \kappa < 1 < \kappa_2''$. *(b)* $\Delta\Pi_O^N > 0$ *if* $\kappa_1''' < \kappa_2''' < 0$; $\Delta\Pi_O^N < 0$ *if* $0 \leq \kappa_2''' \leq 1$ *or* $[0, \kappa_1'''] < \kappa < \kappa_2'''$; *otherwise,* $\Delta\Pi_O^N \geq 0$ *if* $\kappa_2''' \leq \kappa < 1$ *or* $0 \leq \kappa \leq [0, \kappa_1''']$; *for* $\kappa_1''' < 1 < \kappa_2'''$, $\Delta\Pi_O^N < 0$ *if* $0 < \kappa < \kappa_1'''$; *otherwise,* $\Delta\Pi_O^N \geq 0$ *if* $\kappa \geq \kappa_1'''$.

Proposition 6(a) implies the supplier's optimal slotting fee contracts in the presence of the channel competition. The supplier's revenue is too much deprived by the platform when the UR is relatively large; hence, the VFC is more beneficial to the supplier. When the UR is small, adopting the UFC is better for the supplier. The stronger the channel competition is, the more adaptable the UFC is.

Proposition 6(b) describes the platform's optimal slotting fee contracts in the presence of the channel competition. When channel competition is relatively weak, the supplier and platform are in a benign relationship. The UR has a great influence on the formulation of the platform's slotting fee contracts. If the relative gap between the UR and VR is small, the UFC is better for the platform. When the channel conflict is fierce, the VFC becomes an inevitable choice for the platform.

In short, the UFC and VFC both can achieve channel equilibrium when channel competition is weak. On the contrary, under VFC, the supplier and platform are in a cooperative-competitive relationship; hence, they can both obtain additional income through huge market capacity and achieve a win-win situation, as shown in Figure 6. Based on the product category, the platform's VF is generally 1–20% [1]. Facing fierce competition, it can be seen that the VFC is better than the UFC in the dual channel. Therefore, most platforms adopt the VFC in practice.

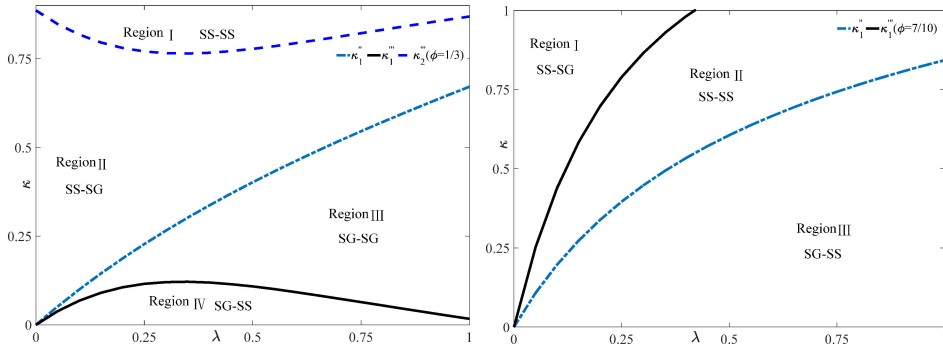

**Figure 6.** Slotting fee contract of the supply chain in format E.

### 5.4. Consumer Surplus

Consumer surplus is the difference between the maximum price consumers are willing to pay for a product and the price they actually pay. Based on this, consumer surplus under different modes can be obtained, as shown in Table 3.

**Table 3.** Consumer surplus under different situations.

| $CS_i^X$ | **S** | **O** | $CS_S^X$ |
|---|---|---|---|
| NG | $\frac{(a-c)^2}{2}$ | N/A | $CS_S^{NG} + CS_O^{NG}$ |
| NS | $\frac{a^2}{2(\lambda+1)^2}$ | N/A | $CS_S^{NS} + CS_O^{NS}$ |
| SG | $\frac{\left(a\left(2-\phi^2+\phi\right)+c\left(\phi^2-2\right)\right)^2}{2\left(2-3\phi^2\right)^2}$ | $\frac{(2a\phi+a-2c\phi)^2}{2(2-3\phi^2)^2}$ | $CS_S^{SG} + CS_O^{SG}$ |
| SS | $\frac{a^2(\phi-2)^2(\phi+1)^2}{2\left(\lambda(\phi^2-2)+3\phi^2-2\right)^2}$ | $\frac{a^2(\lambda+2\phi+1)^2}{2\left(\lambda(\phi^2-2)+3\phi^2-2\right)^2}$ | $CS_S^{SS} + CS_O^{SS}$ |

**Proposition 7.** *Under the SG, the comparison of consumer surplus in different channels:*

*(a) For* $0 \le \phi \le \frac{1}{2}\left(\sqrt{5}-1\right)$, $CS_S^{SG} \ge CS_O^{SG}$ *if* $0 \le \kappa \le \kappa_{CS}^1$; $CS_S^{SG} < CS_O^{SG}$ *if* $\kappa_{CS}^1 < \kappa \le 1$.

*(b) For* $\frac{1}{2}\left(\sqrt{5}-1\right) \le \phi < \sqrt{3}-1$, $CS_S^{SG} < CS_O^{SG}$. *(c) For* $\sqrt{3}-1 < \phi \le \sqrt{\frac{2}{3}}$, $CS_S^{SG} > CS_O^{SG}$.

Proposition 7 shows that consumer surplus in the self-operated channel is always better than that in the agent channel with stronger channel competition, which is not affected by the UR. When channel competition is moderate, consumer surplus is greater in the self-operated channel; otherwise consumer surplus in the agent channel is higher. When channel competition is weaker, consumer surplus is related to the UR. The smaller the UR is, the better consumer surplus in the agent channel is, and vice versa, as shown in Figure 7.

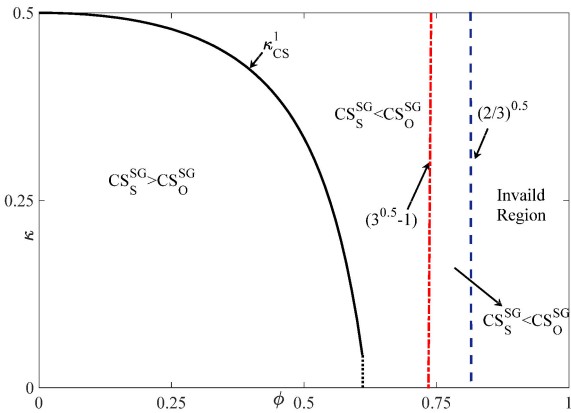

**Figure 7.** Consumer surplus under the SG.

**Proposition 8.** *Under the SS, the comparison of consumer surplus in different channels:*

*(a) For* $0 \le \phi \le \frac{1}{2}\left(\sqrt{5}-1\right)$, *if* $0 \le \lambda \le \lambda_{CS}^1$, $CS_S^{SS} \ge CS_O^{SS}$; *if* $\lambda_{CS}^1 < \lambda \le 1$, $CS_S^{SS} < CS_O^{SS}$.

*(b) For* $\frac{1}{2}\left(\sqrt{5}-1\right) \le \phi < \sqrt{\frac{2}{3}}$, $CS_S^{SS} \ge CS_O^{SS}$. *(c) For* $\sqrt{\frac{2}{3}} < \phi \le 1$, *if* $\lambda \ge \lambda^0$, $CS_S^{SS} < CS_O^{SS}$.

According to Proposition 8, consumer surplus under the SS is similar to that under the SG. When channel competition is minimal, consumer surplus in the different channels is affected by the VR. When the VR is smaller, customers can acquire greater psychological satisfaction in the agent channel, and when the VR is larger, the consumer surplus is better in the self-operated channel. When channel competition is moderate, a higher consumer

surplus is generated in the agent channel. When channel competition is fierce, channel encroachment is not always feasible, therefore, under the dual channel, the self-operated channel can generate a larger customer surplus, as shown in Figure 8.

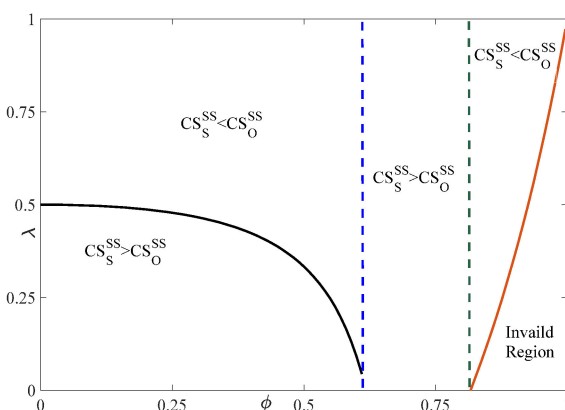

**Figure 8.** Consumer surplus under the SS.

**Proposition 9.** *Under different slotting fee contracts, the comparison of the consumer surplus in agent channel:*

*(a)*$\mathrm{CS}_S^{NG} \geq \mathrm{CS}_S^{NS}$ *if* $0 \leq \kappa \leq \kappa_N^1$; $\mathrm{CS}_S^{NG} < \mathrm{CS}_S^{NS}$ *if* $\kappa_N^1 < \kappa \leq 1$.
*(b)*$\mathrm{CS}_S^{SG} \geq \mathrm{CS}_S^{NG}$; $\mathrm{CS}_S^{SS} \geq \mathrm{CS}_S^{NS}$.

Proposition 9 illustrates that the presence of channel competition always enables consumers to obtain greater consumer surplus in the agent channel. Without the new channel introduction, there is an online agent channel only in the supply chain. When the UR is smaller than the VR, consumer surplus under the UFC is better; otherwise, it is better under the VFC.

**Proposition 10.** *In the presence of the competition, the comparison of the consumer surplus in different slotting fee contracts:*

*(1) In agency channel: (a) For* $0 \leq \phi \leq 2/3$*, if* $0 \leq \lambda \leq \lambda_{CS}^1$, $\mathrm{CS}_S^{SG} \leq \mathrm{CS}_S^{SS}$*; if* $\lambda > \lambda_{CS}^1$, $\mathrm{CS}_S^{SG} > \mathrm{CS}_S^{SS}$ *(b) For* $2/3 < \phi \leq \sqrt{2/3}$*, if* $0 \leq \lambda \leq 1$, $\mathrm{CS}_S^{SG} > \mathrm{CS}_S^{SS}$.

*(2) In self-operated channel: (a) For* $0 \leq \phi \leq 2/3$*, if* $0 \leq \lambda \leq \lambda_{CS}^1$, $\mathrm{CS}_O^{SG} \leq \mathrm{CS}_O^{SS}$*; if* $\lambda > \lambda_{CS}^1$, $\mathrm{CS}_O^{SG} > \mathrm{CS}_O^{SS}$*. (b) For* $2/3 < \phi \leq \sqrt{2/3}$*, if* $0 \leq \lambda \leq 1$, $\mathrm{CS}_O^{SG} > \mathrm{CS}_O^{SS}$.

Proposition 10 illustrates the consumer surplus in the presence of the encroachment. When the channel conflict is serious, regardless of the self-operated channel or the agency channel, consumer surplus under the UFC is better. However, when channel competition is mild, consumer surplus under different slotting fee contracts is affected by the VR. Under the VFC, the smaller the VR is, the larger the consumer surplus in each channel is; otherwise, it is higher under the UFC.

According to the above research, it can be seen that the encroachment can increase consumers' willingness to pay and improve consumer surplus. When channel competition is relatively fierce, slotting fee rates will not have an impact on consumer surplus in different channels. As competition increases, consumers obtain more satisfaction in the self-operated channel than in the agency channel.

**Proposition 11.** $\mathrm{CS}^{SS} \geq \max\left\{\mathrm{CS}^{SG}, \mathrm{CS}^{NG}, \mathrm{CS}^{NG}\right\}$ *if* $1 \leq \kappa \leq \kappa_C^T$*; otherwise,* $\mathrm{CS}^{SG} \geq \max\left\{\mathrm{CS}^{SS}, \mathrm{CS}^{NG}, \mathrm{CS}^{NG}\right\}$.

Proposition 11 shows the total consumer surplus under different cases. From the perspective of the supply chain operation, the mixed channel structure can always obtain higher consumer surplus, as shown in Figure 9. As the variable entry fee model has stronger market response ability, with the improvement of channel competition intensity, price decreases slowly, and consumer surplus increases less. With the intensification of competition, more consumer surplus from self-owned channel to agency channel. When channel competition is fierce, the Mode SS is better for consumer surplus.

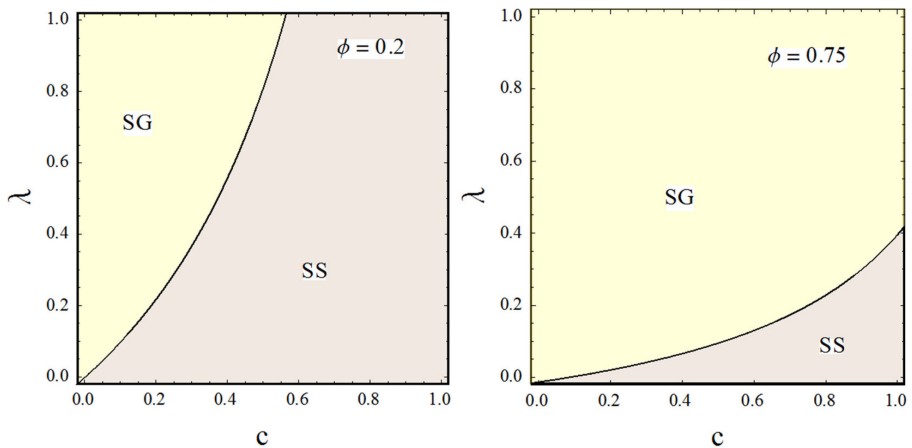

**Figure 9.** Comparative analysis of consumer surplus in the supply chain.

## 6. Extensions

In this section, we relax some constraints to further explore quality decisions and channel configuration strategies in the supply chain. First, we study the endogenous slotting fee under different channel structures, i.e., model E. Second, we consider the consumer quality sensitivity, i.e., model G, and analyze the effect of the quality sensitivity on equilibrium decisions. The associated proofs are provided in Appendix A.

### 6.1. Endogenous Slotting Fee

In Section 4, we consider the exogenous revenue sharing and fixed fee schemes, respectively. Based on Section 4, we extend our study by considering the slotting fee as an endogenous variable, i.e., Model E. According to the profit functions in Section 4, we can obtain the equilibrium results under Modes NG, SG, NS, SS, as shown in Table 4.

**Table 4.** The equilibrium results in Model E.

| D | NG | SG | NS | SS |
|---|---|---|---|---|
| $\widehat{c}^{\,G}\left(\widehat{\lambda}^{\,S}\right)$ | $\frac{a}{2}$ | $\frac{a\left(3\phi^4-3\phi^3-12\phi^2+4\right)}{2\left(3\phi^4-10\phi^2+4\right)}$ | $1$ | $\frac{9\phi^4-16\phi^2-2\phi+4}{3\phi^4-6\phi^3-8\phi^2+6\phi+4}$ |
| $\widehat{w}^{\,Z}$ | N/A | $\frac{a\left(\phi^5+5\phi^4-2\phi^3-12\phi^2+4\right)}{6\phi^4-20\phi^2+8}$ | N/A | $\frac{a\left(6\phi^5+6\phi^4-9\phi^3-9\phi^2+2\phi+2\right)}{9\phi^4-14\phi^2+4}$ |
| $\widehat{Q}_S^{\,Z}$ | $\frac{a}{2}$ | $\frac{a\left(\phi^4-\phi^3-4\phi^2+4\phi+4\right)}{6\phi^4-20\phi^2+8}$ | $\frac{a}{2}$ | $\frac{a\left(3\phi^4-6\phi^3-8\phi^2+6\phi+4\right)}{18\phi^4-28\phi^2+8}$ |
| $\widehat{q}_S^{\,Z}$ | $\frac{a}{2}$ | $\frac{a\left(\phi^4-\phi^3-4\phi^2+4\phi+4\right)}{6\phi^4-20\phi^2+8}$ | $0$ | $\frac{a^2\left(2-6\phi^4-9\phi^3+3\phi^2+8\phi\right)}{18\phi^4-28\phi^2+8}$ |
| $\widehat{Q}_O^{\,Z}$ | N/A | $\frac{a\left(2\phi+2-\phi^3-2\phi^2\right)}{3\phi^4-10\phi^2+4}$ | N/A | $\frac{a(\phi+1)\left(2-3\phi^2\right)}{9\phi^4-14\phi^2+4}$ |
| $\widehat{q}_O^{\,Z}$ | N/A | $\frac{a\left(2\phi+2-\phi^3-2\phi^2\right)}{3\phi^4-10\phi^2+4}$ | N/A | $\frac{a(\phi+1)\left(2-3\phi^2\right)}{9\phi^4-14\phi^2+4}$ |
| $\widehat{\Pi}_S^{\,Z}$ | $\frac{a^2}{8}$ | $\dfrac{a^2\left(\begin{array}{l}48-3\phi^8-30\phi^7-31\phi^6+128\phi^5\\+136\phi^4-168\phi^3-144\phi^2+64\phi\end{array}\right)}{8\left(3\phi^4-10\phi^2+4\right)^2}$ | $0$ | $\frac{a^2\left(2-6\phi^4-9\phi^3+3\phi^2+8\phi\right)}{18\phi^4-28\phi^2+8}$ |
| $\widehat{\Pi}_O^{\,Z}$ | $\frac{a^2}{4}$ | $\frac{a^2\left(\phi^4-2\phi^3-3\phi^2+8\phi+6\right)}{4\left(3\phi^4-10\phi^2+4\right)}-T$ | $\frac{a^2}{4}$ | $\frac{a^2\left(9\phi^4-12\phi^2+4\phi+6\right)}{4\left(9\phi^4-14\phi^2+4\right)}-T$ |

**Proposition 12.** *(a)* ① $\widehat{q}_S^{SG} > \max\left\{\widehat{q}_S^{NG}, \widehat{q}_S^{NS}, \widehat{q}_S^{SS}\right\}$ *if* $0 < \phi < \phi_E^1$; *otherwise,* $\widehat{q}_S^{SG} \leq$ $\max\left\{\widehat{q}_S^{NG}, \widehat{q}_S^{NS}, \widehat{q}_S^{SS}\right\}$. ② $\widehat{q}_O^{SS} > \widehat{q}_O^{SG}$.

Proposition 12 and Figure 10 jointly show that the quality investment strategies under Model E. Compared with a single channel structure, the emergence of channel competition can always promote suppliers to invest in quality, so as to obtain higher quality to respond to market changes. Meanwhile, Mode SG is more beneficial to make a quality investment for the supplier when the competition intensity is low; otherwise, the supplier has a stronger investment under Mode SS. For platforms, there is always a higher quality investment in the SS scenario. Because the supplier's decisions are affected by wholesale contracts and slotting fee contracts under Mode SG, and the quality decision changes little with the competition. However, the quality is affected by a single factor and becomes more sensitive to market changes under Mode SS.

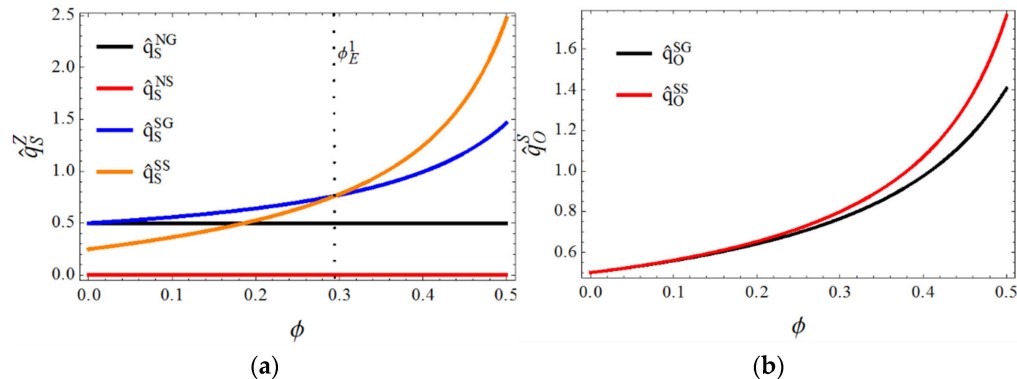

**Figure 10.** The quality level of the supplier and platform under Model E. (**a**) The supplier (**b**) the platform.

**Proposition 13.** *(a)* $\widehat{\prod}_S^{SG} > \max\left\{\widehat{\prod}_S^{NG}, \widehat{\prod}_S^{NS}, \widehat{\prod}_S^{SS}\right\}$ *if* $0 < \phi < \phi_E^2$; *otherwise,* $\widehat{\prod}_S^{SS} >$ $\max\left\{\widehat{\prod}_S^{NG}, \widehat{\prod}_S^{NS}, \widehat{\prod}_S^{SG}\right\}$. *(b)* $\widehat{\prod}_O^{NG}\left(\widehat{\prod}_O^{NS}\right) > \left\{\widehat{\prod}_O^{SG}, \widehat{\prod}_O^{SS}\right\}$ *if* $0 < T < T_E^1$; $\widehat{\prod}_O^{SG} >$ $\left\{\widehat{\prod}_O^{NG}\left(\widehat{\prod}_O^{NS}\right), \widehat{\prod}_O^{SS}\right\}$ *if* $T \geq T_E^1$.

Proposition 13 shows the Mode preference of the supplier and platform. The supplier always prefers Mode SG when channel competition is low; otherwise, Mode SG always can always promote supplier profitability, as shown in Figure 11a. That is because a new channel introduction always can increase sales, so the supplier always can profit under the dual-channel structure. Meanwhile, under Mode SS, the supplier can obtain higher profit through wholesale price premiums with the mild competition. However, the platform's Mode preference is affected by channel competition and introduction cost, as shown in Figure 11b. Mode SG is beneficial to the platform when cost efficiency $(T/\phi)$ is low; otherwise, the platform always prefers to sell products by single-channel structure.

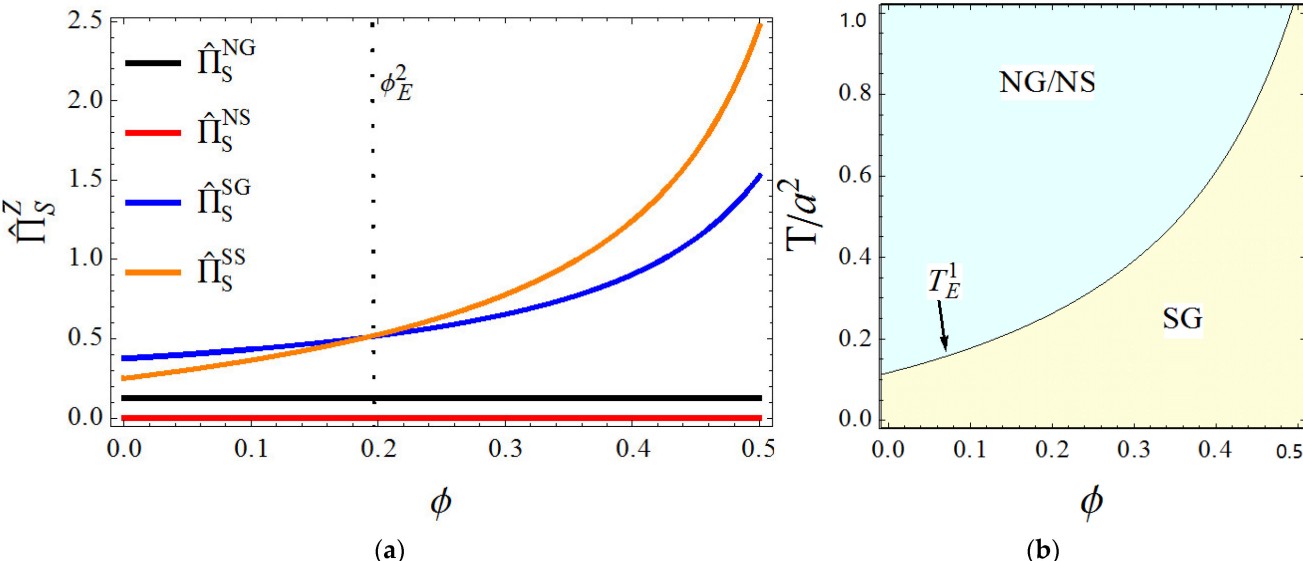

(a)

(b)

**Figure 11.** Mode preference of supply chain under Model E. (**a**) Mode preference of the supplier (**b**) Mode preference of the platform.

*6.2. Consumer Quality Sensitivity*

In Section 4, we consider that consumers have the same sensitivity to prices and qualities [46]. We extend the study and explore the effect of dynamic quality sensitivity $\theta$ on demand, i.e., Model G. Therefore, in the absence of the channel introduction, the inverse demand function is as follows: $p_i^N = a + \theta q_i - Q_i$ [48]. In the presence of the channel introduction, the inverse demand function is as follows: $p_i^S = a + \theta q_i - Q_i - \phi Q_j$ [24,48]. Referring to the profit functions in Section 4, we can obtain the equilibrium results under Modes NG, SG, NS, SS, as shown in Table 5.

**Observation 1.** *Figures 12 and 13 show the supplier's and platform's quality investment strategies, considering consumer quality sensitivities. Compared with Propositions 1 and 2, it can be seen that the supplier and platform need higher costs to improve the same quality efficiency when the quality sensitivity of consumers is low. For the supplier and platform, because the marginalization among members can be reduced in Mode SS, they have stronger advantages and higher quality investment willingness. On the contrary, a fixed slotting fee contract has a stronger advantage and can stimulate suppliers to invest in quality when consumers are highly sensitive to quality.*

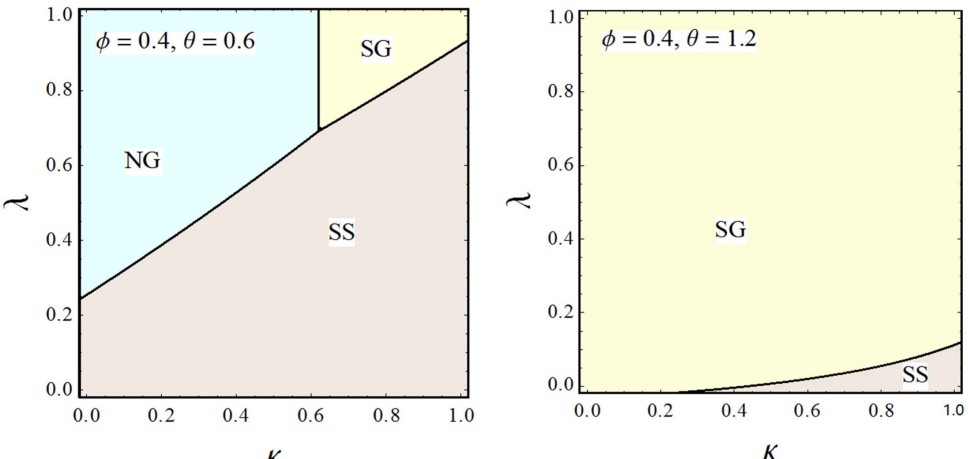

**Figure 12.** The quality strategy of the supplier under Model G.

**Table 5.** The equilibrium results in Model G.

| D. | NG | SG | NS | SS |
|---|---|---|---|---|
| $\widetilde{w}^Z$ | N/A | $\frac{a\left(\theta^6-6\theta^4-2\theta^2\left(\phi^2-6\right)+\phi^3+4\phi^2-8\right)-c\phi^3}{\left(\theta^2-2\right)\left(2\theta^4-8\theta^2-3\phi^2+8\right)}$ | N/A | $\dfrac{a\left(\begin{array}{l}\theta^6(\lambda-1)^2-2\theta^2\left(2\lambda^2\phi-\lambda\left(\phi^2+4\phi-4\right)+\phi^2-6\right)+4\phi^2\\+2\theta^4(\lambda-1)(\lambda(\phi-1)+3)+\lambda^2\phi^3+2\lambda\phi\left(\phi^2-4\right)+\phi^3-8\end{array}\right)}{\left(\theta^2(\lambda-1)+2\right)\left(2\theta^4(\lambda-1)-4\theta^2(\lambda-2)+(\lambda+3)\phi^2-8\right)}$ |
| $\widetilde{Q}_S^Z$ | $\frac{a-c}{\theta^2-2\theta+2}$ | $\frac{a\left(\theta^2(\phi+8)-2\theta^4+\phi^2-2\phi-8\right)+c\left(2\theta^4-8\theta^2-\phi^2+8\right)}{\left(\theta^2-2\right)\left(2\theta^4-8\theta^2-3\phi^2+8\right)}$ | $\frac{a\theta(1-\lambda)}{\theta^2\lambda-\theta^2+2}$ | $\frac{a\left(2\theta^4(\lambda-1)+\theta^2(-\lambda(\phi+4)+\phi+8)+(\lambda+1)\phi^2-2\phi-8\right)}{\left(\theta^2(\lambda-1)+2\right)\left(2\theta^4(\lambda-1)-4\theta^2(\lambda-2)+(\lambda+3)\phi^2-8\right)}$ |
| $\widetilde{q}_S^Z$ | $\frac{\theta(a-c)}{\theta^2-2\theta+2}$ | $\frac{\theta\left(a\left(\theta^2(\phi+8)-2\theta^4+\phi^2-2\phi-8\right)+c\left(2\theta^4-8\theta^2-\phi^2+8\right)\right)}{\left(\theta^2-2\right)\left(2\theta^4-8\theta^2-3\phi^2+8\right)}$ | $\frac{a}{\theta^2\lambda-\theta^2+2}$ | $\frac{a\theta(\lambda-1)\left(2\theta^4(\lambda-1)+\theta^2(-\lambda(\phi+4)+\phi+8)+(\lambda+1)\phi^2-2\phi-8\right)}{\left(\theta^2(1-\lambda)-2\right)\left(2\theta^4(\lambda-1)-4\theta^2(\lambda-2)+(\lambda+3)\phi^2-8\right)}$ |
| $\widetilde{Q}_O^Z$ | N/A | $\frac{a\left(2-\theta^2+2\phi\right)-2c\phi}{2\theta^4-8\theta^2-3\phi^2+8}$ | N/A | $\frac{a\theta\left(\theta^2(\lambda-1)+2(\phi+1)\right)}{2\theta^4(1-\lambda)-4\theta^2(2-\lambda)-(\lambda+3)\phi^2+8}$ |
| $\widetilde{q}_O^Z$ | N/A | $\frac{\theta\left(a\left(2-\theta^2+2\phi\right)-2c\phi\right)}{2\theta^4-8\theta^2-3\phi^2+8}$ | N/A | $\frac{a\left(\theta^2(\lambda-1)+2(\phi+1)\right)}{2\theta^4(1-\lambda)-4\theta^2(2-\lambda)-(\lambda+3)\phi^2+8}$ |
| $\widetilde{\Pi}_S^Z$ | $\frac{(a-c)^2}{2\left(\theta^2-2\theta+2\right)}$ | $\dfrac{\left(\begin{array}{l}a^2\left(3\theta^4-4\theta^2(\phi+3)+\phi^2+8\phi+12\right)+c^2\left(\phi^2+8\right)-\\2ac\left(2\theta^4-2\theta^2(\phi+4)+\phi^2+4\phi+8\right)+c^2\left(2\theta^4-8\theta^2\right)\end{array}\right)}{2\left(2-\theta^2\right)\left(2\theta^4-8\theta^2-3\phi^2+8\right)}$ | $\frac{a^2(1-\lambda)}{2\theta^2(\lambda-1)+4}$ | $\frac{a^2(\phi+1)(\lambda(\phi-1)+\phi+3)}{2\lambda\left(2-\phi^2\right)-6\phi^2+4}$ |
| $\widetilde{\Pi}_O^Z$ | $\frac{c(a-c)}{\theta^2-2\theta+2}$ | $\dfrac{\left(\begin{array}{l}a^2\left(\theta^2-2\right)^2\left(\theta^2-2(\phi+1)\right)^2+2ac\left(4\theta^8-32\theta^6-12\theta^4\left(\phi^2-8\right)\right)\\+2ac\left(3\phi^4-6\phi^3-48\phi^2+64+\theta^2\left(3\phi^3+48\phi^2-128\right)\right)-128c^2-\\2c^2\left(4\theta^8-32\theta^6+\theta^4\left(96-10\phi^2\right)+8\theta^2\left(5\phi^2-16\right)+3\phi^4-40\phi^2\right)\end{array}\right)}{2\left(2-\theta^2\right)\left(2\theta^4-8\theta^2-3\phi^2+8\right)^2}-T$ | $\frac{a^2\lambda\left(\theta^2(\lambda-1)-\theta\lambda+\theta+1\right)}{\left(\theta^2(\lambda-1)+2\right)^2}$ | $\dfrac{a^2\left(\begin{array}{l}\lambda^2\left(2\phi^3-2\phi^2-4\phi+1\right)+(2\phi+1)^2\\+2\lambda\left(3\phi^4-3\phi^3-8\phi^2+4\phi+5\right)\end{array}\right)}{2\left(\lambda\left(\phi^2-2\right)+3\phi^2-2\right)^2}-T$ |

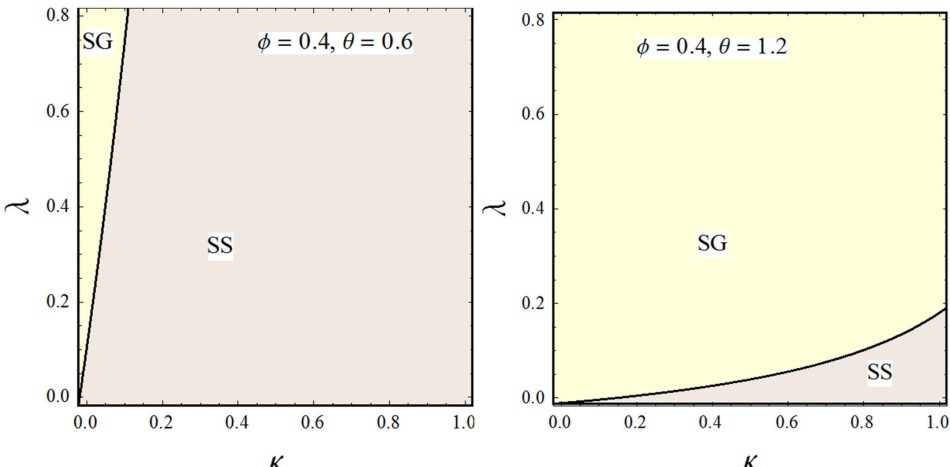

**Figure 13.** The quality strategy of the platform under Model G.

**Observation 2.** *Figures 14 and 15 show the Mode preference of the supplier and the platform. For the supplier, no matter how consumer quality sensitivity changes, the dual-channel structure is always more beneficial than the single-channel structure. Meanwhile, with the increase in consumer quality sensitivity, the advantages of SG Mode expand and SS Mode gradually is cannibalized. From the platform perspective, the competitive market environment deteriorates when consumer quality sensitivity is low; the platform prefers NS Mode with a higher VF. With the increase in quality sensitivity of consumers, the advantages of dual-channel operation are improved and the platform always prefers SG or SS Mode. When the VF is high, the platform prefers SS; otherwise, SG is better. In a word, the increase in consumer sensitivity can promote dual-channel operation and realize Pareto improvement of supply chain members.*

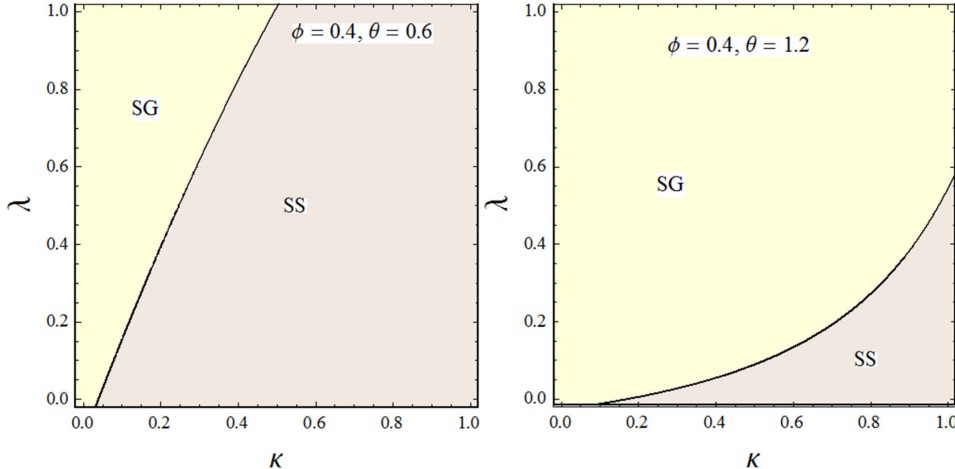

**Figure 14.** Mode preference of the supplier.

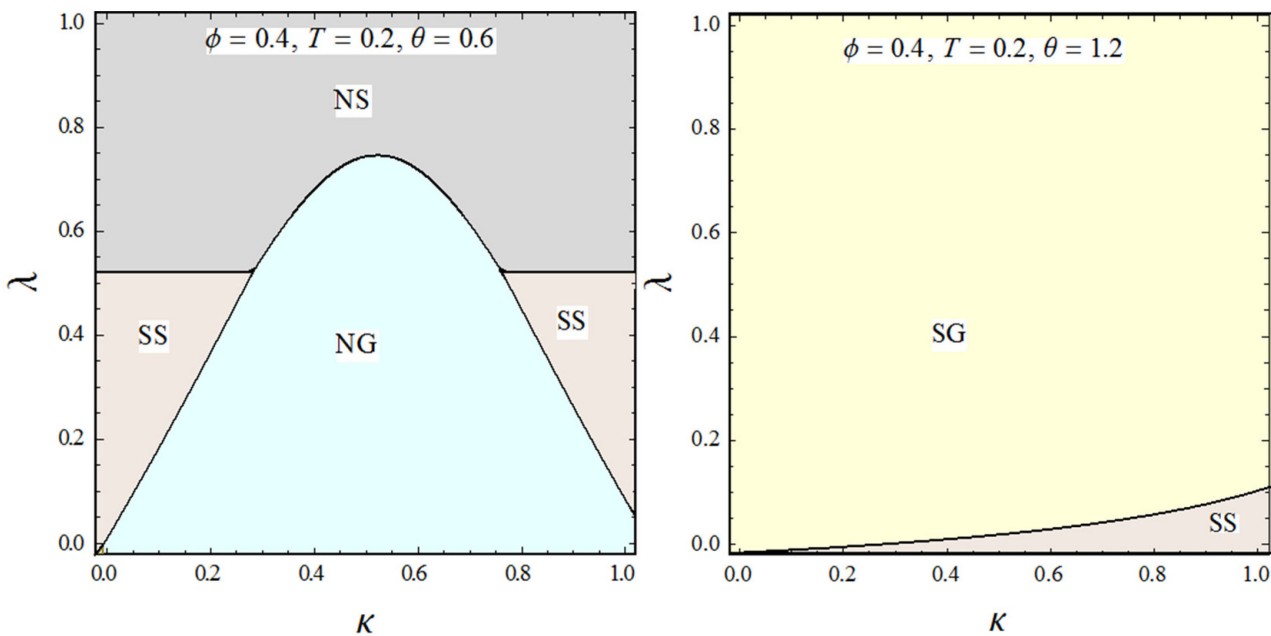

**Figure 15.** Mode preference of the platform.

## 7. Conclusions

In the digital economy era, the platform plays an important role in optimizing resources, promoting industrial upgrading, expanding the market and maintaining sustainable business operations. With increasingly fierce competition, the platform encroaches to gain a dominant position. In the paper, we consider a supply chain composed of a supplier and a platform to investigate the impact of the platform's encroachment under different slotting fee contracts on the decisions and profits, which provides a theoretical framework and guidance for enterprises' sustainable operations.

### 7.1. Summary Findings

In the platform supply chain, this paper explores the impact of channel competition and slotting fee rates on the decisions and profits of the supply chain. Furthermore, through numerical analysis, the slotting fee contract design of the supply chain, quality investment and channel selection strategy are discussed. The main findings are as follows:

Firstly, within a certain range of channel competition, the establishment of a self-operated channel can improve quality and increase the incomes of suppliers and platforms. Therefore, some large platforms (e.g., Tmall.com) have established a self-operated channel, which can effectively stimulate suppliers to invest in quality, resulting in maintaining the platform's sustainability and corporate social responsibility. However, due to the lack of an effective supervision mechanism, the increase in slotting fees weakens quality investment willingness, which is not conducive to the benign operation of the platform (e.g., Pinduoduo.com, accessed on 16 October 2022). In practice, the quality supervision mechanism of the large-scale platform is not relatively perfect, which leads to uneven quality. However, as consumers become more sensitive to quality, supply chain members have to improve quality investment to meet the market so as to maintain a sustainable operation.

Secondly, there are multiple interactions between the platform and the supplier. Once the platform increases the slotting fee cost, the supplier will make full use of its power to protect its interests by maintaining the wholesale price and reducing quality investment. As a result, consumer satisfaction varies with the change of channel structure and contract mode, so it is very important for the platform to formulate a reasonable slotting fee contract.

Lastly, the channel introduction is a platform-led strategy. When the channel introduction cost is not too large, the dual-channel structure always benefits the platform.

Meanwhile, the channel introduction can increase consumer surplus, which contributes to expanding the market and enhancing brand influence. When channel competition is fierce, the UFC leads to the degradation of the direct sales channel and damages the interests of the supplier. While under the VFC, the relationship between the supplier and platform becomes cooperative-competitive, which alleviates the competition and realizes a win-win outcome. When the slotting fee is endogenous, suppliers always prefer SG Mode when channel competition is mild; otherwise, suppliers always prefer SS Mode. However, the SS Mode is always inferior to the platform.

### 7.2. Theoretical Implications

Through mathematical derivation and model analysis, we have obtained the following theoretical implications: the findings of this work provide some theoretical contributions to the existing literature with respect to strategy choice and sustainable operations of the platform supply chain. Compared to the existing literature [11,14,19], we highlight the following three important theoretical contributions.

Firstly, based on the study of supply chain operations under endogenous and exogenous slotting fees, we identify the factors influencing the design of slotting fee contracts between suppliers and platforms, i.e., the intensity of channel competition and slotting fee rates under different covenants. Furthermore, the results present the marginal conditions of the conversion between UFC and VFC, which filled the gap in the prior literature [4,11].

Secondly, different from the previous literature on upstream supply chain encroachment [32,33], this work implies that platform encroachment strategies are related to encroachment cost and platform market scale. Superior platforms (i.e., large market scale and low encroachment costs) tend to adopt a mixed channel strategy. Meanwhile, other members of the supply chain can always benefit from the encroachment, which has not been found in previous studies [19,46]. Therefore, this study is a significant contribution to the literature on supply chain encroachment.

Thirdly, contrary to the results of previous literatures [11,14], this work proves that channel intrusion has a positive impact on quality investment. Thus, our research offers insights into the investment extent and influence factors of quality in different channels, including investment level, consumer strategic behavior, contract design and member conflicts of interest. In addition, it provides guidance for further understanding of the motivations that influence platforms and suppliers to invest in quality.

### 7.3. Managerial Implications

According to the strategic analysis of the supplier and platform, we put forward the following managerial implications, which provide some suggestions for firms to make operational decisions in the competitive market.

Firstly, from the perspective of platforms, the channel introduction strategy of platforms is mainly affected by its own operational costs. If platforms are mature and have a large market capacity, such as JD.com and Amazon.com, accessed on 16 October 2022, it is beneficial to introduce a self-operated channel. If platforms are in the initial stage of development, such as Pinduoduo.com, accessed on 7 October 2022, they contribute to operating through an agent channel structure, which can avoid multiple risks in operation, quickly open the market and improve market coverage.

Secondly, from the suppliers' point of view, when suppliers sell products only through agent channels, they are restricted by the slotting fee of the platform. Instead, suppliers have more choices and improve their position in the channel through premium wholesale prices when the platform opens up new channels, so suppliers always prefer to sell products through differentiated mixed channels.

Thirdly, from the perspective of the supply chain, whether the supply chain design is UFV or VFC in the context of monopolistic operation, which can always achieve balance and maintain stable operation. With the fierce online competition, the roles of firms in the

supply chain are gradually diversified. In a competitive environment, the VFC is more conducive to sustainable operation.

In addition, the results provide guidance for the department of management. On the one hand, the government should take some incentive measures to promote competition among channels so as to ensure the high service quality and product quality of online platforms. On the other hand, the government should make out a standardized supervision system for online platforms to prevent the emergence of vicious and unhealthy competition because the emergence of channel competition can drive up prices.

*7.4. Further Research*

By taking the proportional slotting fee, quality investment and channel competition into account, our research might guide the sustainable operation of the platform under different business modes. Future work on this topic can be considered to establish an effective supervision mechanism to ensure product quality investment. In addition, consumer strategic behavior, information disclosure and supply chain coordination mechanisms under competitive channels can be explored in the future.

**Author Contributions:** Conceptualization, C.L. and P.X.; methodology, C.L.; software, C.L.; validation, C.L. and Y.L.; formal analysis, C.L.; investigation, Y.L.; resources, P.X.; data duration, P.X.; writing—original draft preparation, C.L.; writing—review and editing, Y.L. and P.X.; visualization, Y.L. and P.X.; supervision, C.L.; project administration, P.X.; funding acquisition, Y.L. All authors have read and agreed to the published version of the manuscript.

**Funding:** This work was financially supported by Shenyang Philosophy and Social Science Special Fund (Grant No. SY202204ZC).

**Institutional Review Board Statement:** Not applicable.

**Informed Consent Statement:** Not applicable.

**Data Availability Statement:** No new data were created or analyzed in this study.

**Conflicts of Interest:** The authors declare no conflict of interest.

**Appendix A**

**Proof of Lemma 1.** According to the backward induction, the decisions are obtained. Firstly, because $\frac{\partial^2 \Pi_S^{NG}}{\partial q_S^2} = -1 < 0$, $\Pi_S^{NG}$ is strictly concave with respect to $q_S^{NG}$. According to $\frac{\partial \Pi_S^{NG}}{\partial q_S} = 0$, quality investment level of the supplier is $q_S = Q_S$. Secondly, because $\frac{\partial^2 \Pi_S^{NG}}{\partial Q_S^2} = -1 < 0$, $\Pi_S^{NG}$ is strictly concave with respect to $Q_S^{NG}$. According to $\frac{\partial \Pi_S^{NG}}{\partial Q_S} = 0$, the supplier optimal sales volume $Q_S^{NG}$. When platform does not encroach, by backward induction the optimal decisions and profits of the supplier and platform can be obtained under the UFC. □

**Proof of Corollary 1.** The proof of Proposition 1 is simple, and thus omitted. □

**Proof of Lemma 2.** Firstly, because $\frac{\partial^2 \Pi_S^{SG}}{\partial q_S^2} = -1 < 0$ and $\frac{\partial^2 \Pi_O^{SG}}{\partial q_O^2} = -1 < 0$. According to $\frac{\partial \Pi_S^{SG}}{\partial q_S} = 0$ and $\frac{\partial \Pi_O^{SG}}{\partial q_O} = 0$, quality investment level of the supplier is $q_S(Q_S, Q_O, w)$ and quality investment level of the platform is $q_O(Q_S, Q_O, w)$. Secondly, because $\frac{\partial^2 \Pi_S^{SG}}{\partial Q_S^2} = -1 < 0$ and $\frac{\partial^2 \Pi_O^{SG}}{\partial Q_O^2} = -1 < 0$, according to $Q_S(w)$ and $Q_O(w)$. Thirdly, because $\frac{\partial^2 \Pi_S^{SG}}{\partial w^2} = \frac{3\phi^2 - 2}{(\phi^2 - 1)^2}$, if $0 \le \phi^2 \le \frac{2}{3}$ and $\frac{\partial^2 \Pi_S^{SG}}{\partial w^2} \le 0$, according to $\frac{\partial \Pi_S^{SG}}{\partial w} = 0$, the wholesale price of the supplier is $w^{SG}$. Then we reverse into $Q_S^{SG}$, $Q_O^{SG}$, $q_S^{SG}$ and $q_O^{SG}$, the optimal decision and equilibrium

profits of the supplier and the platform are obtained under the UFC, when the platform chooses to encroach. □

**Proof of Corollary 2.** According to the existence, non-negative and uniqueness of the equilibrium solution, we can obtain known $0 \leq \phi^2 \leq \frac{2}{3}$ and $a > c$. Taking the first derivative of decisions with respect to $\phi$ or $c$ under different modes.

① $\frac{\partial Q_S^{SG}}{\partial \phi} = \frac{\partial q_S^{SG}}{\partial \phi} = \frac{a(3\phi^2+8\phi+2)-8c\phi}{(2-3\phi^2)^2}$, because $\frac{a(3\phi^2+8\phi+2)}{8\phi} > c$, $\frac{\partial Q_S^{SG}}{\partial \phi} = \frac{\partial q_S^{SG}}{\partial \phi} > 0$; $\frac{\partial Q_S^{SG}}{\partial c} = \frac{\partial q_S^{SG}}{\partial c} = \frac{\phi^2-2}{2-3\phi^2} < 0$.

② $\frac{\partial w^{SG}}{\partial \phi} = \frac{\phi(a(3\phi^3-6\phi-2)-3c\phi(\phi^2-2))}{(2-3\phi^2)^2}$, because, $\frac{a(3\phi^3-6\phi-2)}{3\phi(\phi^2-2)} > c$, $\frac{\partial w^{SG}}{\partial \phi} > 0$; $\frac{\partial w^{SG}}{\partial c} = \frac{\phi^3}{2-3\phi^2} > 0$.

③ $\frac{\partial Q_O^{SG}}{\partial \phi} = \frac{\partial q_O^{SG}}{\partial \phi} = \frac{a(6\phi^2+6\phi+4)-2c(3\phi^2+2)}{(2-3\phi^2)^2}$, because $\frac{a(3\phi^2+3\phi+2)}{3\phi^2+2} > c$, $\frac{\partial Q_O^{SG}}{\partial \phi} = \frac{\partial q_O^{SG}}{\partial \phi} > 0$; $\frac{\partial Q_O^{SG}}{\partial c} = \frac{\partial q_O^{SG}}{\partial c} = -\frac{2\phi}{2-3\phi^2} < 0$.

④ $\frac{\partial \Pi_S^{SG}}{\partial c} = -\frac{2c(\phi^2+2)-2a(\phi^2+2\phi+2)}{6\phi^2-4}$, because $\frac{a(\phi^2+2\phi+2)}{\phi^2+2} > c$, $\frac{\partial \Pi_S^{SG}}{\partial c} < 0$; $\frac{\partial \Pi_S^{SG}}{\partial \phi} = \frac{a^2(6\phi^2+11\phi+4)-2ac(3\phi^2+8\phi+2)+8c^2\phi}{(2-3\phi^2)^2}$, because $\frac{1}{4}(3a\phi+4a) > c$, $\frac{\partial \Pi_S^{SG}}{\partial \phi} > 0$.

⑤ $\frac{\partial \Pi_O^{SG}}{\partial \phi} = \frac{3ac\phi^2(3\phi^2+16\phi+6)-2a^2(6\phi^3+9\phi^2+7\phi+2)+4c^2\phi(2-9\phi^2)}{(3\phi^2-2)^3}$, $\frac{\partial \Pi_O^{SG}}{\partial \phi} < 0$. $\frac{\partial \Pi_O^{SG}}{\partial c} = \frac{(2a(3\phi^4-3\phi^3-12\phi^2+4)-4c(3\phi^4-10\phi^2+4))}{2(2-3\phi^2)^2}$. If $0 < \phi < 0.56$ and $\kappa > \kappa_{SG}^S$, $\frac{\partial \Pi_O^{SG}}{\partial c} > 0$; if $0 < \phi < 0.56$ and $0 \leq \kappa \leq \kappa_{SG}^S$, $\frac{\partial \Pi_O^{SG}}{\partial c} > 0$; if $0.56 < \phi < \phi_{SG}^C$, $\frac{\partial \Pi_O^{SG}}{\partial c} < 0$; otherwise, $\phi_{SG}^C \leq \phi \leq \phi_{SG}$, $\frac{\partial \Pi_O^{SG}}{\partial c} > 0$. Where $\phi_{SG}^C = \sqrt{\frac{1}{3}\left(5 - \sqrt{13}\right)}$, $\phi_{SG} = \sqrt{\frac{2}{3}}$, $\kappa_{SG}^S = \frac{2c(3\phi^4-10\phi^2+4)}{3\phi^4-3\phi^3-12\phi^2+4}$. □

**Proof of Lemma 3.** Firstly, because $\frac{\partial^2 \Pi_S^{NS}}{\partial q_S^2} = -1$, according to $\frac{\partial \Pi_S^{NS}}{\partial q_S^2} = 0$, quality investment level of the supplier is $q_S(Q_S)$. Secondly, because $\frac{\partial^2 \Pi_S^{NS}}{\partial Q_S^2} = \lambda^2 - 1 < 0$, according to $\frac{\partial \Pi_S^{NS}}{\partial Q_S} = 0$, optimal sales volume of the supplier is $Q_m$. By backward induction, the optimal decision and equilibrium profits of the supplier and the platform are obtained under the VFC, when the platform does not choose to encroach. □

**Proof of Corollary 3.** Taking the first derivative of decisions with respect to $\lambda$ under different modes.

These are easy to be proved as follows: $\frac{\partial Q_S^{NS}}{\partial \lambda} = -\frac{a}{(\lambda+1)^2} < 0$; $\frac{\partial q_S^{NS}}{\partial \lambda} = -\frac{2a}{(\lambda+1)^2} < 0$. $\frac{\partial \Pi_S^{NS}}{\partial \lambda} = -\frac{a^2}{(\lambda+1)^2} < 0$; $\frac{\partial \Pi_O^{NS}}{\partial \lambda} = \frac{a^2(1-\lambda)}{(\lambda+1)^3} > 0$. □

**Proof of Lemma 4.** Firstly, because $\frac{\partial^2 \Pi_S^{SS}}{\partial q_S^2} = -1 < 0$ and $\frac{\partial^2 \Pi_O^{SS}}{\partial q_O^2} = -1 < 0$, according to $\frac{\partial \Pi_S^{SS}}{\partial q_S} = 0$ and $\frac{\partial \Pi_O^{SS}}{\partial q_O} = 0$, quality investment level of the supplier and the platform are $q_S(Q_S, Q_O, w)$ and $q_O(Q_S, Q_O, w)$.

Secondly, because $\frac{\partial^2 \Pi_S^{SS}}{\partial Q_S^2} = \lambda^2 - 1$ and $\frac{\partial^2 \Pi_O^{SS}}{\partial Q_O^2} = -1$, according to $\frac{\partial \Pi_S^{SS}}{\partial Q_S} = 0$ and $\frac{\partial \Pi_O^{SS}}{\partial Q_O} = 0$, optimal sales volume of the supplier and platform are $Q_S^{SS}(w)$ and $Q_O^{SS}(w)$.

Thirdly, because $\frac{\partial^2 \Pi_S^{SS}}{\partial w^2} = \frac{\lambda(\phi^2-2)+3\phi^2-2}{(\lambda+1)(\phi^2-1)^2}$, if $\lambda \geq \lambda^{SS}$, $\frac{\partial^2 \Pi_S^{SS}}{\partial w^2} \leq 0$, where $\lambda^{SS} = \frac{2-3\phi^2}{\phi^2-2}$.

According to $\frac{\partial \Pi_S^{SS}}{\partial w} = 0$, wholesale price of the supplier is $w^{SS}$. By backward induction, the optimal decision and equilibrium profits of the supplier and the platform are obtained under the VFC, when the platform chooses to encroach. □

**Proof of Corollary 4.** Taking the first derivative of decisions with respect to $\phi$ or $\lambda$ under different modes.

① $\frac{\partial Q_S^{SS}}{\partial \phi} = \frac{a(\lambda(\phi^2+2)+3\phi^2+8\phi+2)}{(\lambda(\phi^2-2)+3\phi^2-2)^2} > 0;$ $\frac{\partial Q_S^{SS}}{\partial \lambda} = \frac{a(2-\phi)(\phi+1)(\phi^2-2)}{((\lambda+3)\phi^2-2(\lambda+1))^2} < 0.$ $\frac{\partial q_S^{SS}}{\partial \phi} = \frac{a(1-\lambda)(\lambda(\phi^2+2)+3\phi^2+8\phi+2)}{(\lambda(\phi^2-2)+3\phi^2-2)^2} > 0;$ $\frac{\partial q_S^{SS}}{\partial \lambda} = -\frac{4a(\phi-2)(\phi-1)(\phi+1)^2}{(\lambda(\phi^2-2)+3\phi^2-2)^2} < 0.$

② $\frac{\partial w^{SS}}{\partial \phi} = \frac{a\Lambda_1}{(\lambda(\phi^2-2)+3\phi^2-2)^2}.$ If $0 < \lambda < g(\phi)$, $\frac{\partial w^{SS}}{\partial \phi} < 0$, if $g(\phi) \leq \lambda < 1$, $\frac{\partial w^{SS}}{\partial \phi} \geq 0;$ $\frac{\partial w^{SS}}{\partial \lambda} = \frac{2a\phi(\phi+1)^2(\phi^2-3\phi+2)}{(\lambda(\phi^2-2)+3\phi^2-2)^2} > 0.$

③ $\frac{\partial Q_O^{SS}}{\partial \phi} = \frac{\partial q_O^{SS}}{\partial \phi} = \frac{2a\Lambda_2}{(\lambda(\phi^2-2)+3\phi^2-2)^2} > 0;$ $\frac{\partial Q_O^{SS}}{\partial \lambda} = \frac{\partial q_O^{SS}}{\partial \lambda} = \frac{2a\phi(\phi^2-\phi-2)}{(\lambda(\phi^2-2)+3\phi^2-2)^2} < 0.$

④ $\frac{\partial \pi_m^{SS}}{\partial \phi} = \frac{a^2\Lambda_3}{(\lambda(\phi^2-2)+3\phi^2-2)^2} > 0$ $\frac{\partial \pi_m^{SS}}{\partial \lambda} = \frac{a^2(\phi-2)^2(\phi+1)^2}{(\lambda(\phi^2-2)+3\phi^2-2)^2} < 0.$ $\frac{\partial \pi_o^{SS}}{\partial \phi} = \frac{a^2\Lambda_4}{(\lambda(2-\phi^2)+2-3\phi^2)^3} > 0.$

⑤ $\frac{\partial \pi_o^{SS}}{\partial \lambda} = \frac{a^2(2-\phi^2+\phi)\Lambda_5}{(\lambda(\phi^2-2)+3\phi^2-2)^3}.$ For $\phi \in [0, 0.467]$, if $0 < \lambda < \lambda^1$, $\frac{\partial \pi_o^{SS}}{\partial \lambda} > 0$; if $\lambda^1 < \lambda < 1$, $\frac{\partial \pi_o^{SS}}{\partial \lambda} < 0.$ For $\phi \in [0.467, \sqrt{\frac{2}{3}}]$, $\frac{\partial \pi_o^{SS}}{\partial \lambda} < 0.$

Where $\Lambda_1 = \begin{pmatrix} \lambda^2(\phi^4-4\phi^2+2\phi+4)-2\phi+ \\ \lambda(4\phi^4-6\phi^2+4)+\phi(3\phi^3-6\phi) \end{pmatrix}$, $\Lambda_2 = \begin{pmatrix} \lambda^2\phi+3\phi^2+3\phi+2 \\ +\lambda(\phi^2+4\phi+2) \end{pmatrix}$,

$\Lambda_3 = \begin{pmatrix} \lambda^2\phi+6\phi^2+11\phi+4 \\ +2\lambda(\phi^2+2\phi+2) \end{pmatrix}$, $\Lambda_3 = \begin{pmatrix} \lambda^2\phi+6\phi^2+11\phi+4 \\ +2\lambda(\phi^2+2\phi+2) \end{pmatrix}$,

$\Lambda_5 = \begin{pmatrix} \lambda(3\phi^4-6\phi^3-8\phi^2+6\phi+4) \\ -9\phi^4+16\phi^2+2\phi-4 \end{pmatrix}$, $g(\phi) = \lambda^0 = \frac{\Lambda_6-2-2\phi^4+3\phi^2}{\phi^4-4\phi^2+2\phi+4}$,

$\Lambda_6 = \sqrt{\begin{pmatrix} \phi^8+6\phi^6-4\phi^5-19\phi^4+ \\ 4\phi^3+16\phi^2+8\phi+4 \end{pmatrix}}$, $\lambda^1 = \frac{4-2\phi-16\phi^2+9\phi^4}{4+6\phi-8\phi^2-6\phi^3+3\phi^4}.$ $\square$

**Proof of Proposition 1.** ① Note $q_S^S = q_S^{SG} - q_S^{SS} = \frac{4a\lambda(\phi+1)^2(\phi^2-3\phi+2)-c(2-\phi^2)Y_1}{Y_1Y_2}.$ By solving the $q_S^D = 0$, we can obtain $\kappa = \kappa_s^1$. For $0 \leq \lambda \leq \lambda_s^1$ and $0 \leq \kappa_s^1 \leq 1$, if $0 \leq \kappa \leq \kappa_s^1$, $q_S^{SG} \geq q_S^{SS}$; if $\kappa_s^1 \leq \kappa \leq 1$, $q_S^{SG} < q_S^{SS}$. For $\lambda_s^1 < \lambda \leq 1$, $\kappa_s^1 > 1$, hence $q_S^{SG} < q_S^{SS}$.

② Note $q_S^N = q_S^{NG} - q_S^{NS} = \frac{\phi(2a\phi+a-2c\phi)}{Y_2}.$ By solving the $q_S^N = 0$, we can get $k = \kappa_s^0$. Hence, if $0 \leq \kappa \leq \kappa_s^0$, $q_S^{NS} \leq q_S^{NG}$; $\kappa_s^0 < \kappa \leq 1$, $q_S^{NS} > q_S^{NG}$.

Where $Y_1 = \begin{pmatrix} 4a\lambda(\phi+1)^2(\phi^2-3\phi+2)- \\ c(\phi^2-2)(\lambda(\phi^2-2)+3\phi^2-2) \end{pmatrix}$, $Y_2 = (\lambda(\phi^2-2)+3\phi^2-2)$, $\kappa_s^0 = \frac{2\lambda}{\lambda+1}$; $\kappa_s^1 = \frac{4\lambda(1+\phi)^2(2-3\phi+\phi^2)}{(\phi^2-2)(3\phi^2+\lambda\phi^2-2-2\lambda)}$; $\lambda_s^1 = \frac{4-8\phi^2+3\phi^4}{4+4\phi-8\phi^2-4\phi^3+3\phi^4}.$ $\square$

**Proof of Proposition 2.** Note $q_O^S = q_O^{SG} - q_O^{SS} = \frac{2\phi(a\lambda Y_3-cY_1)}{(2-3\phi^2)Y_1}.$ By solving the $q_O^S = 0$, we can obtain $\kappa = \kappa_o^1$. For $\lambda \leq \lambda_o^1$ and $0 \leq \kappa_o^1 \leq 1$; if $0 \leq \kappa \leq \kappa_o^1$, $q_O^{SG} \geq q_O^{SS}$; if $\kappa_o^1 < \kappa \leq 1$, $q_O^{SG} < q_O^{SS}$. For $\lambda_o^1 < \lambda \leq 1$ and $q_O^{SG} < q_O^{SS}$. Where $\kappa_o^1 = \frac{a\lambda(\phi^2-\phi-2)}{\lambda(\phi^2-2)+3\phi^2-2}$; $\lambda_o^1 = \frac{2-3\phi^2}{\phi}$, $Y_3 = (a\lambda(2-\phi^2+\phi)+c(\lambda(\phi^2-2)+3\phi^2-2)).$ $\square$

**Proof of Proposition 3.** Note $\Delta\Pi_S^G = \Pi_S^{SG} - \Pi_S^{NG} = \frac{(2a\phi+a-2c\phi)^2}{2Y_2} > 0.$

Note $\Delta\Pi_O^G = \Pi_O^{SG} - \Pi_O^{NG} = \frac{Y_5}{2(2-3\phi^2)^2}.$ By solving the $T = T_G$, we can obtain $\Delta\Pi_O^G = 0.$ Where $Y_5 = \begin{pmatrix} (2a\phi+a)^2-6ac(2\phi+1)\phi^3- \\ 2(2c^2(1-3\phi^2)\phi^2+T(2-3\phi^2)^2) \end{pmatrix}$, $T_G = \frac{(2a\phi+a)^2-6ac(2\phi+1)\phi^3+4c^2(3\phi^2-1)\phi^2}{2(2-3\phi^2)^2} > 0.$ $\square$

**Proof of Proposition 4.** Note $\Delta\Pi_S^S = \Pi_S^{SS} - \Pi_S^{NS} = \frac{a^2(\lambda+2\phi+1)^2}{2(\lambda+1)(2-3\phi^2-\lambda(\phi^2-2))} > 0.$

Note $\Delta\Pi_O^S = \Pi_O^{SS} - \Pi_O^{NS} = \frac{a^2 Y_6 - 2(\lambda+1)^2 T Y_2^2}{2(\lambda+1)^2 Y_2^2}$. By solving the $\Delta\Pi_O^S = 0$, we can obtain

$T = T_S$. Where $Y_6 = \begin{pmatrix} \lambda^4\left(2\phi^3 - 2\phi^2 - 4\phi + 1\right) + 2\lambda^3\left(2\phi^4 - \phi^3 - 6\phi^2 + 2\right) + (2\phi+1)^2 \\ +2\lambda^2\left(-5\phi^3 + \phi^2 + 8\phi + 3\right) - 2\lambda\left(6\phi^4 + 3\phi^3 - 8\phi^2 - 8\phi - 2\right) \end{pmatrix}$,

$T_S = \frac{a^2 Y_7}{2(\lambda+1)^2 Y_2^2}$, $Y_7 = \begin{pmatrix} \lambda^4\left(2\phi^3 - 2\phi^2 - 4\phi + 1\right) + 2\lambda^3\left(2\phi^4 - \phi^3 - 6\phi^2 + 2\right) + (2\phi+1)^2 \\ +2\lambda^2\left(-5\phi^3 + \phi^2 + 8\phi + 3\right) - 2\lambda\left(6\phi^4 + 3\phi^3 - 8\phi^2 - 8\phi - 2\right) \end{pmatrix}$.

$\square$

**Proof of Proposition 5.** (a) Note $\Delta\Pi_S^N = \Pi_S^{NS} - \Pi_S^{NG} = ac - \frac{a^2\lambda}{\lambda+1} - \frac{c^2}{2}$. From $\Delta\Pi_S^N = 0$, we can obtain $\kappa = \kappa_1$ or $\kappa = \kappa_2$. Because $0 \le \kappa_1 \le 1$, $1 \le \kappa_1 \le 2$. Hence, if $\kappa_1 < \kappa \le 1$, $\Delta\Pi_S^N > 0$; if $0 < \kappa \le \kappa_1$, $\Delta\Pi_S^N \le 0$.

(b) Note $\Delta\Pi_O^N = \Pi_O^{NS} - \Pi_O^{NG} = \frac{a^2\lambda}{(\lambda+1)^2} - ac + c^2$. From $\Delta\Pi_O^N$, we can obtain $\kappa = \kappa_1'$ or $\kappa = \kappa_2'$, and $0 \le \kappa_1' \le \kappa_2' < 1$. Hence, if $\kappa_1' \le \kappa \le \kappa_2'$, $\Delta\Pi_O^N < 0$; if $0 \le \kappa < \kappa_1'$ or $\kappa_2' < \kappa \le 1$, $\Delta\Pi_O^N > 0$. Where $\kappa_1 = \frac{\lambda+1-\sqrt{1-\lambda^2}}{\lambda+1}$ and $\kappa_2 = \frac{\lambda+1+\sqrt{1-\lambda^2}}{\lambda+1}$, $\kappa_1' = \frac{\lambda}{\lambda+1}$ and $\kappa_2' = \frac{1}{\lambda+1}$. $\square$

**Proof of Proposition 6.** Note $\Delta\Pi_S^V = \Pi_S^{SS} - \Pi_S^{SG} = \frac{Y_8}{2(3\phi^2-2)Y_2}$. By solving the $\Delta\Pi_S^V = 0$, we can obtain $\kappa = \kappa_1''$ and $\kappa = \kappa_2''$. Where $\kappa_1'' = -\frac{2\left(\phi^2+2\phi+2\right)Y_2 + 4\left(3\phi^2-2\right)\sqrt{Y_9}}{2\left(4-\lambda\phi^4+4\lambda-3\phi^4-4\phi^2\right)}$; $\kappa_1'' = \frac{4\left(3\phi^2-2\right)\sqrt{Y_9} - 2\left(\phi^2+2\phi+2\right)Y_2}{2\left(4-\lambda\phi^4+4\lambda-3\phi^4-4\phi^2\right)}$, $Y_8 = \begin{pmatrix} 2a^2\lambda\left(\phi^2 - \phi - 2\right)^2 - 2ac\left(\phi^2 + 2\phi + 2\right)Y_2 \\ +c^2\left(\lambda\left(\phi^4 - 4\right) + 3\phi^4 + 4\phi^2 - 4\right) \end{pmatrix}$,

$Y_9 = \begin{pmatrix} \left(2 + 2\phi + \phi^2\right)^2\left(3\phi^2 - 2\right) + \lambda^2\left(8 - 8\phi^2 + \phi^6\right) \\ +2\lambda\phi\left(-8 - 14\phi + 5\phi^3 + 2\phi^4 + 2\phi^5\right) \end{pmatrix}$.

Note $\Delta\Pi_O^V = \Pi_O^{SS} - \Pi_O^{SG} = \frac{a^2\lambda\left(\phi^2 - \phi - 2\right)Y_{10} - G}{(2-3\phi^2)^2 Y_2^2}$. By solving the $\Delta\Pi_O^V = 0$, we can obtain $\kappa = \kappa_1'''$ and $\kappa = \kappa_2'''$. Where $Y_{10} = \begin{pmatrix} 9\lambda\phi^5 - 2(\lambda + 33)\phi^4 - 2(8\lambda + 3)\phi^3 \\ +4(\lambda + 11)\phi^2 + (8\lambda + 4)\phi + 27\phi^6 - 8 \end{pmatrix}$,

$Y_{11} = \lambda\begin{pmatrix} 3\phi^6 - 3\phi^5 - 18\phi^4 \\ +6\phi^3 + 28\phi^2 - 8 \end{pmatrix}$, $\kappa_1''' = \frac{\left(Y_{11} - \left(3\phi^2 - 2\right)\left(\sqrt{Y_{12}} - 3\phi^4 + 3\phi^3 + 12\phi^2 - 4\right)\right)}{2\left(3\phi^4 - 10\phi^2 + 4\right)Y_2}$,

$\kappa_2''' = \frac{\left(Y_{11} + \left(3\phi^2 - 2\right)\left(\sqrt{Y_{12}} - 3\phi^4 + 3\phi^3 + 12\phi^2 - 4\right)\right)}{2\left(3\phi^4 - 10\phi^2 + 4\right)Y_2}$, $G = Y_2^2\begin{pmatrix} ac\left(3\phi^4 - 3\phi^3 - 12\phi^2 + 4\right) \\ -c^2\left(3\phi^4 - 10\phi^2 + 4\right) \end{pmatrix}$,

$Y_{12} = \begin{pmatrix} \lambda^2\left(\phi^8 - 14\phi^7 + 5\phi^6 + 80\phi^5 - 24\phi^4 - 136\phi^3 + 64\phi + 16\right) + \left(3\phi^4 - 3\phi^3 - 12\phi^2 + 4\right)^2 \\ +\lambda\left(24\phi^7 - 30\phi^8 + 182\phi^6 - 88\phi^5 - 336\phi^4 + 64\phi^3 + 208\phi^2 - 32\right) \end{pmatrix}$.

$\square$

**Proof of Proposition 7.** Note $CS_S^S = CS_S^{SG} - CS_O^{SG} = \frac{Y_{13}}{2(2-3\phi^2)^2}$. By solving the $CS_S^S = 0$, we can obtain $\kappa = \kappa_{CS}^1$. For $0 \le \phi \le \sqrt{3} - 1$, $\phi^4 - 8\phi^2 + 4 \ge 0$; if $0 \le \phi \le \frac{1}{2}\left(\sqrt{5} - 1\right)$, $\kappa_{CS}^1 \le 1$, hence $0 \le \kappa \le \kappa_{CS}^1$, $CS_S^{SG} \ge CS_O^{SG}$; otherwise, $\kappa_{CS}^1 < \kappa \le 1$, $CS_S^{SG} < CS_O^{SG}$, if $\frac{1}{2}\left(\sqrt{5} - 1\right) \le \phi < \sqrt{3} - 1$, $\kappa_{CS}^1 > 1$, $CS_S^{SG} < CS_O^{SG}$. For $\sqrt{3} - 1 < \phi \le \sqrt{\frac{2}{3}}$, $\phi^4 - 8\phi^2 + 4 < 0$, $CS_S^{SG} > CS_O^{SG}$. Where $\kappa_{CS}^1 = \frac{\left(\phi^2 + \phi - 1\right)}{\phi^2 + 2\phi - 2}$; $\lambda_{CS}^1 = 1 - \phi - \phi^2$; $\lambda^{SS} = \frac{2 - 3\phi^2}{\phi^2 - 2}$, $Y_{13} = \begin{pmatrix} a^2\left(\phi^4 - 2\phi^3 - 7\phi^2 + 3\right) + c^2\left(\phi^4 - 8\phi^2 + 4\right) \\ -2ac\left(\phi^4 - \phi^3 - 8\phi^2 + 4\right) \end{pmatrix}$. $\square$

The proof of Propositions 8, 9 and 10 are similar to that of Proposition 6, and thus omitted.

**Proof of Proposition 11.** Let $CS^{SG} - CS^{NG} = \frac{a^2 Y_{12}}{2(2-3\phi^2)^2} > 0$. $CS^{SS} - CS^{NS} = \frac{a^2 Y_{13}}{2(\lambda+1)^2 Y_2^2} > 0$. $CS^{SG} - CS^{SS} = \frac{a^2 Y_{14}}{2Y_1^2 Y_2^2}$; when $\kappa = \kappa_C^T$, $CS^{SG} - CS^{SS} = 0$. Where $\kappa_C^T = \frac{\lambda\left(\phi^2 - \phi - 2\right)}{Y_2}$,

$$Y_{12} = \begin{pmatrix} (1 - 8\phi^4 - 2\phi^3 + 13\phi^2 + 8\phi) + 12\phi^2 \\ + 2k\phi(8\phi^3 + \phi^2 - 12\phi - 4) - 4\phi^4 \end{pmatrix}, Y_{14} = \begin{pmatrix} \lambda(\phi^6 - \phi^5 - 2\phi^4 + 12\phi^3 + 4\phi^2 - 12\phi - 8) + \\ 2(3\phi^6 - 3\phi^5 - 2\phi^4 + 14\phi^3 + 12\phi^2 - 8\phi - 8) \end{pmatrix},$$

$$Y_{13} = \begin{pmatrix} \lambda^4 + 4\lambda^3(\phi + 1) + \lambda^2(6 - 2\phi^3 + 5\phi^2 + 16\phi) + 8\phi + 1 \\ \lambda(4 - 4\phi^4 - 4\phi^3 + 18\phi^2 + 20\phi) - 8\phi^4 - 2\phi^3 + 13\phi^2 \end{pmatrix}, Y_{14} = \begin{pmatrix} \lambda Y_{14}(\phi^2 - \phi) + \kappa^2(\phi^4 + 4)Y_2^2 - \\ 2\kappa(\phi^4 - \phi^3 + 4\phi + 4)Y_2^2 - 2\lambda Y_{14} \end{pmatrix}. \square$$

The proof of Propositions 12 and 13 is simple, and thus omitted.

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
