# Peer review of "Sustainable Strategy Analysis: Platform Channel Configuration and Slotting Fee Design under Differentiated Quality Investment"

_sustainability, doi:10.3390/su142316095_

Round 1
Reviewer 1 Report
Please see the appendix.

Author Response
Dear Editor,
We sincerely thank the reviewer for their comments, the modifications to the manuscript are presented in the attached document, please download and check.
Yours sincerely,
Chunyu Li, Peng Xing, Yanting Li

Reviewer 2 Report
Dear Authors,
Thank you for your interesting paper.
Two suggestions to improve the paper.
First, the Introduction is long, cut the words and be precise!
Second, It is recommended to elaborate on the implications of your findings to the governance of digital platforms from a policy perspective.
Best,
Reviewer
Author Response
Dear Editor,
We sincerely thank the reviewer for the comments, the modifications to the manuscript are presented in the attached document, please download and check.
Yours sincerely,
Chunyu Li, Peng Xing, Yanting Li

Reviewer 3 Report
This paper considers a supply chain composed of a supplier and a platform and investigates the platform’s optimal strategies, i.e., pricing, quality investment, channel format, and slotting fee contract. The paper examines an issue of considerable interest and significance. This paper has a certain degree of innovation.
1. The main objectives of the research are defined at the introduction of the study. The authors described the study problem and research questions, the importance of the study, and the hypotheses as well.
2. All the tables and figures are clear, understandable, and relevant, and sources are indicated in each case well.
3. Conclusions are suitable for gaining new results and initiating further or new research.
Overall, I think it is well written and worth publication and I agree to publish this paper by Sustainability.
Author Response
Dear reviewer,
We would like to take this opportunity to thank the Reviewers for their insightful and helpful comments on our manuscript, which enabled us have a comprehensive understanding of this paper. Thank you very much for your time and consideration.
Yours sincerely,
Chunyu Li, Peng Xing, Yanting Li